# **Stripe Patterns in Wind Forecasts Induced by Physics-Dynamics**

# 2 Coupling on a Staggered Grid in CMA-GFS 3.0

- 3 Jiong Chen<sup>1,2,3</sup>, Yong Su<sup>1,2,3</sup>, Zhe Li<sup>1,2,3</sup>, Zhanshan Ma<sup>1,2,3</sup>, Xueshun Shen<sup>1,2,3</sup>
- <sup>1</sup>State Key Laboratory of Severe Weather Meteorological Science and Technology, Beijing, 10081,
- 5 China

- 6 <sup>2</sup>CMA Earth System Modeling and Prediction Centre (CEMC), Beijing, 10081, China
- 7 <sup>3</sup>Key Laboratory of Earth System Modeling and Prediction, China Meteorological Administration,
- 8 Beijing, 10081, China
- 9 Correspondence to: Jiong Chen (cjiong@cma.gov.cn) and Xueshun Shen (shenxs@cma.gov.cn)
  - Abstract. An unphysical stripe pattern is identified in low-level wind field in China Meteorological Administration Global Forecast System (CMA-GFS), characterized by meridional stripes in ucomponent and zonal stripes in v-component. This stripe noise is primarily confined to the planetary boundary layer over land. The structural mismatch between static field variations and the observed  $2\Delta x$  noise amplitude suggests that locally forced mechanisms from surface inhomogeneity alone cannot explain the wind stripe patterns. Meanwhile, pure dynamical core simulations exhibit no such noise, confirming that the dynamical core itself does not generate these patterns. These results suggest that staggered-grid mismatch in physics-dynamics coupling is likely the primary mechanism. Idealized two-dimensional experiments demonstrate that combining one-dimensional dynamic-core advection and physics-based vertical diffusion on a staggered grid generates  $2\Delta x$ wavelength spurious waves when surface friction is non-uniform. One-dimensional linear wave analysis further confirms that staggered-grid coupling between dynamic advection and inhomogeneous damping forcing induces dispersion errors in wave solutions. Sensitivity tests validate that eliminating grid mismatch in physics-dynamics coupling removes this stripe noise. These findings collectively indicate that while staggered grids benefit the dynamic core's numerical stability and accuracy, their inherent grid mismatch with physics parameterizations requires specialized coupling strategies to avoid spurious noise. Potential solutions to remedy this issue are discussed.

#### 1. Introduction

The wind field, as a fundamental state variable in atmospheric dynamics, governs not only the accuracy of circulation forecasts but also controls the spatiotemporal distributions of temperature, humidity, aerosols, and cloud microphysical properties through advection processes. Numerical weather prediction (NWP) systems simulate three-dimensional wind field evolution through numerical integration of atmospheric governing equations with multi-scale physical parameterizations. The momentum equations describe wind field evolution through three key forces: the pressure gradient force, Coriolis force, and nonlinear advection terms. The numerical discretization of advection terms is particularly important as it directly affects solution stability (Durran, 2010). Parameterized physical processes at subgrid scale significantly influence wind field predictions. Surface stress not only exerts substantial control over the wind intensity in both the near-surface and planetary boundary layers (PBL) through turbulent momentum transport (Blackadar, 1957; Stull, 1988), but also significantly influences the location of the surface westerlies and upper-level jets in the midlatitudes (Robinson, 1997; Chen et al., 2007). When flow encounters subgrid-scale orography, it excites gravity waves that propagate vertically, altering large-scale winds through momentum flux divergence (McFarlane, 1987; Kim et al., 2003; Kim and Doyle, 2005). Mesoscale topographic blocking and turbulent form drag modify low-level wind fields, which in turn affect global circulation patterns (Beljaars et al., 2004; Sandu et al., 2016). In addition to dynamics and physical processes, the consistency of dynamic-physics coupling is equally critical for overall model performance (Bauer et al., 2015; Gross et al., 2016; 2018). In NWP models, numerical noise typically stems from non-physical high-frequency oscillations induced by dynamical process discretization (Wurtele, 1961; Arakawa, 1966; Mesinger & Arakawa, 1976). Additionally, spatial grid mismatch in physics-dynamic coupling can generate computational noise (Chen et al., 2020). While consistent spatial grids between dynamical and physical processes are theoretically preferable for seamless coupling, practical implementations often employ differing horizontal and vertical grid configurations. For vertical discretization, two conventional grid arrangements are employed: (1) the Lorenz grid (L-grid; Lorenz, 1960), which collocates thermodynamic variables (e.g., potential temperature and moisture) and horizontal wind components at the half-levels (mass levels), with the vertical velocity and geopotential typically

located at the full-levels (interface levels); and (2) the Charney-Phillips grid (CP-grid; Charney & Phillips, 1957), which staggers the thermodynamic variables at the full-levels above and below the corresponding half-levels of the horizontal wind, while the vertical velocity is also defined at the full-levels. Chen et al. (2017) demonstrated that Lorenz-grid physics coupled with CP-grid dynamics generates non-physical computational modes in PBL scheme solutions. Chen et al. (2020) confirmed that unified grid coupling significantly improves the stratocumulus representation and overall forecast skill while eliminating this spurious noise. These findings underscore that dynamic-physical coupling inconsistencies not only introduce numerical noise but also substantially degrade model performance.

In global models, physical processes are typically computed in single columns, independent of horizontal grid configuration. Thus, horizontally distributed noise cannot originate from physical parameterizations if the underlying static physical data is noise-free. In two-dimensional rectangular grids, horizontal grid configurations are categorized into non-staggered (A-grid) and staggered (B grid to E grid) types, distinguished by their arrangement of wind components (u, v) and mass variables (height z or pressure p) (Arakawa and Lamb, 1977). Among these, the Arakawa C-grid – where mass variables are collocated at grid centers while velocity components are staggered at cell interfaces - exhibits optimal accuracy when the grid spacing resolves scales smaller than the Rossby radius of deformation (Batteen & Han, 1981; Xu & Lin, 1993; Randall, 1994). This superiority stems from its inherent conservation properties and reduced numerical dispersion in simulating geostrophic adjustment processes (Arakawa and Lamb, 1977; Randall, 1994). The Arakawa-C grid has been widely implemented as a preferred discretization framework in structured-grid NWP systems, including the Weather Research and Forecasting (WRF) Model's Advanced Research core (Skamarock et al., 2008) and operational systems such as the UK Met Office Unified Model (Walters et al., 2017), the NOAA Global Forecast System (GFS) (Sela, 2010), the Environment and Climate Change Canada (ECCC) Global Environmental Multiscale (GEM) model (Côté et al., 1998), and the China Meteorological Administration's (CMA) operational GRAPES system (Chen et al., 2008). Figure 1 illustrates the wind field distribution on an Arakawa C-grid, where the physicsparameterized winds are located at the central mass points while the dynamically computed winds are staggered at the cell interfaces. This inherent wind field grid mismatch is typically handled via two-step interpolation during physics-dynamics coupling. However, as demonstrated by Chen et al.

(2020), the grid inconsistency during physics-dynamics coupling introduces spurious computational noise.

High-order horizontal diffusion (Xue, 2000) and filters (Shapiro, 1970, 1975; Raymond and Garder, 1988) are standard techniques for suppressing high-frequency computational noise in NWP models. These approaches were successfully implemented in the GRAPES model and effectively mitigated small-scale noise (Wang et al., 2008). Compared to CMA-GFS 2.4 (originally GRAPES\_GFS V2.4), the new CMA-GFS 3.0 has removed fourth-order horizontal diffusion to improve small-scale system simulation capabilities (Shen et al., 2023). Following the upgrade to CMA-GFS 3.0, stripe patterns became evident in the low-level wind fields—an issue absent in previous versions.

This study systematically investigates the underlying causes of these stripe patterns in CMA-GFS 3.0, particularly examining physics-dynamics coupling mismatches as the probable source. Section 2 details physics-dynamics coupling scheme on a staggered grid for the wind field prediction in CMA-GFS 3.0 and characterizes the observed horizontal wind noise patterns. Section 3 identifies noise sources via idealized experiments and analytical wave solutions, revealing fundamental discretization constraints in physics-dynamics coupling. Section 4 designs sensitivity experiments to verify the origin of horizontal wind noise in CMA-GFS 3.0, conclusively attributing it to physics-dynamics coupling mismatches. Conclusions and discussions are presented in Section 5.

## 2. The wind prediction in the PBL in CMA GFS

#### 2.1 Model description and physics-dynamic coupling in wind prediction

The CMA-GFS (formerly known as GRAPES\_GFS) was developed based on a two-time-level semi-implicit semi-Lagrangian (SISL) non-hydrostatic dynamic core. Its development began in 2007, and the system became operational in late 2015 (Shen et al., 2020). The model employs a horizontal Arakawa C-grid on a regular latitude-longitude grid, combined with a vertical CP grid using a height-based terrain-following coordinate system (Chen et al., 2008).

Physics package includes the Rapid Radiative Transfer Model for GCMs (RRTMG) for

longwave and shortwave radiation (Pincus et al. 2003; Morcrette et al. 2008), the double-moment

microphysics and an explicit prognostic cloud cover scheme (Ma et al, 2018), the common land model (CoLM) (Dai et al. 2003), the subgrid-scale orographic (SSO) scheme including gravity wave drag (GWD) (Chen et al. 2016) and turbulence orographic form drag (TOFD) (Beljaar et al., 2004), the new Simplified Arakawa Schubert (NSAS) convection, and the new medium-range forecast (NMRF) for vertical diffusion (Han and Pan 2011). The implementation of vertical diffusion and cloud physics schemes on the CP grid eliminated computational noise induced by physics-dynamics coupling mismatches, leading to significant improvements in stratocumulus cloud simulations and overall forecast skill scores (Chen et al., 2020).

Figure 1. Wind configuration in Arakawa C grid; the subscript 'p' denotes the physics grid points

In CMA-GFS's dynamical core, horizontal winds  $u_{i\pm\frac{1}{2},j}$  and  $v_{i,j\pm\frac{1}{2}}$  are staggered at cell interfaces, while physics-parameterized winds  $u_p$  and  $v_p$  are collocated at mass center (Figure 1). In our coupling scheme, linear interpolation bridges the grid mismatch for physics-dynamics coupling:

$$P(u_{i,j}) = P\left(\frac{u_{i-\frac{1}{2},j} + u_{i+\frac{1}{2},j}}{2}\right)$$

$$P(v_{i,j}) = P\left(\frac{v_{i,j-\frac{1}{2}} + v_{i,j+\frac{1}{2}}}{2}\right)$$
(1)

- where P denotes the physics parameterization operator. When feeding physics tendencies back to
- the dynamical core, the coupling follows:

$$\Delta u_{i+\frac{1}{2},j}^{P} = \frac{1}{2} [P(u_{i,j}) + P(u_{i+1,j})] \Delta t$$

$$\Delta v_{i,j+\frac{1}{2}}^{P} = \frac{1}{2} [P(v_{i,j}) + P(v_{i,j+1})] \Delta t$$
(2)

- where  $\Delta V^P$  denotes the physics-induced wind increment to be fed back to the dynamic core.
- However, such interpolation-based coupling schemes often fail to mitigate computational noise
- induced by physics-dynamics grid mismatches (Chen et al. 2020).
- As theoretically anticipated, lower-level winds of CMA-GFS 3.0 exhibit distinct stripe patterns.
- We therefore systematically investigate these anomalies in detail, first characterizing their
- spatiotemporal structures, and then evaluating potential links to physics-dynamics coupling
- mechanisms.

#### 2.2 Spatiotemporal distribution of wind forecast noise: a case study

- In this case study—hereafter referred to as Exp\_Ctrl—we utilize CMA-GFS 3.0 with a
- horizontal resolution of 0.25° (approximately 25 km at the equator) and 87 vertical levels extending
- up to 1 hPa. The simulation spans from 1200 UTC 1 July 2021, initialized with European Centre
- for Medium-Range Weather Forecasts (ECMWF) ERA5 reanalysis data (Hersbach et al., 2020; data
- available at https://cds.climate.copernicus.eu/datasets/). Hourly forecast fields are saved directly on
- the model's Arakawa C-grid at model levels throughout the 5-day simulation.
- Global distribution of the wind field noise reveals pronounced land-sea contrasts, with
- predominant stripe patterns over land while relatively smooth over oceans. Figure 2 displays the 18-
- hour forecast (valid at 0600 UTC) of near-surface winds at the lowest model level over South Asia
- (5-34°N, 88-113°E), a region encompassing both land and ocean. Distinct stripe patterns dominate
- the wind field over land areas, where the u-component displays meridional stripes and the v-
- component exhibits zonal stripes. With the model's 0.25° resolution, these structures are clearly
- resolved with a wavelength of  $0.5^{\circ}$  ( $2\Delta x$ ), demonstrating characteristic  $2\Delta x$  wave properties.

Figure 2. 18-hour forecast of (a) u- and (b) v- components at the lowest model level from CMA-GFS 3.0

Figure 3 displays vertical cross-sections of u- and v- components along 30°N and 100°E, respectively, for the region shown in Fig. 2. Pronounced  $2\Delta x$  wave patterns are evident throughout the PBL in both components, whereas the wind field becomes markedly smoother in the free atmosphere above the PBL. The strongest  $2\Delta x$  signatures are concentrated in the bottom of the PBL.

Figure 3 Vertical distribution of 18-hour forecasted winds: (a) *u*-component along 30°N; (b) *v*-component along 100°E. Black solid lines denote the PBL height.

Figure 4 presents the 120-hour (5-day) evolution of surface-layer u- and v- components along the zonal and meridional transects shown in Fig. 3. The  $2\Delta x$  oscillations exhibit continuous presence throughout the period. Their amplitude shows strong diurnal variations but no systematic growth with forecast days, suggesting that this noise is unlikely to directly induce model instability. This persistent noise pattern shows daytime intensification and nighttime weakening, suggests a potential linkage to the inherent diurnal cycle of the development of PBL.

Figure 4. Temporal evolution of winds at the lowest model level (0-120 hours): (a) u-component along 30°N, (b) v-component along 100°E.

The vertical structure of low-level winds and their diurnal variations in the surface layer imply a connection between the stripe patterns and momentum-related physical processes, particularly turbulent transport within the PBL. However, the single-column design of physical schemes permits PBL diffusion to affect such horizontal wind noise solely via surface friction forcing. In the following section, we systematically examine the mechanisms for these  $2\Delta x$  stripe patterns.

### 3. Tracing the cause of wind stripes predicted in CMA-GFS 3.0

#### 3.1 Examination of associated surface static fields

The surface static fields directly governing surface friction in the wind field are surface inhomogeneity descriptors, principally characterized by subgrid orographic variability and aerodynamic roughness length. Figure 5 displays the spatial distributions of subgrid orography standard deviation and momentum roughness length across the domain shown in Fig. 2.

Figure 5. Horizontal distributions of (a) Standard deviation of sub-grid orography; (b) Roughness length over the study region (domain same as Fig. 2).

The distributions of the two static fields in Fig. 5 differ markedly from the wind stripe patterns in Fig. 2, indicating that surface parameter inhomogeneity is not the direct origin of the stripe noise. However, visual comparison with Fig. 2 demonstrates a distinct spatial correspondence between the stripe noise locations and the surface roughness distribution: the  $2\Delta x$  noise patterns predominantly occur over areas with higher roughness, while remaining absent over smooth surfaces. The Andaman Islands (11–13°N, 93°E), located between the Bay of Bengal and the Andaman Sea, generate localized surface inhomogeneity in this predominantly oceanic region. Consistent with this inhomogeneity, Fig. 2 exhibits obvious meridional stripes in the u- component precisely over this archipelago. These observations imply that although the surface inhomogeneity parameters themselves exhibit no stripe patterns—and thus cannot directly produce the wind stripes through physical processes—the spatial organization of  $2\Delta x$  noise systematically correlates with surface inhomogeneity.

#### 3.2 Dynamical Core Tests: Noise Assessment in Physics-off simulation

Section 3.1 demonstrated that the surface-friction-related static fields lack horizontal noise

patterns (Fig. 5), definitively excluding surface-mediated physical processes—which are computed in single columns—as possible sources of the observed  $2\Delta x$  wind stripes. We now evaluate the dynamical core's potential contribution through a physics-free simulation using identical configuration described in section 2.2. Figure 6 displays 18-hour forecasts of near-surface u- and v-components over the same domain as in Fig. 2. The absence of surface friction yields stronger winds compared to the default physics-on simulation (Fig. 2), and more critically, the wind fields exhibit smooth distributions completely free of  $2\Delta x$  stripes. This definitive evidence excludes the dynamical core as the source of the observed low-level wind noise.

Figure 6. Same as Fig. 2, but for the physics-off experiment

Having definitively excluded both the dynamical core and physical parameterizations as direct sources of these horizontal noise patterns, the residual evidence points to the physics-dynamics coupling interface as the sole remaining candidate. As schematized in Fig. 1, CMA-GFS employs staggered grids between physics and dynamics components, requiring linear interpolation for wind field coupling. This inherent grid mismatch is now the most probable mechanism generating the observed  $2\Delta x$  stripe noise. We proceed with idealized experiments and theoretical analysis to

examine whether the grid mismatch between physics and dynamics could induce the computational noise.

#### 3.3 Two-dimensional idealized test

227

- The governing equation for the idealized simulation combining one-dimensional horizontal advection with vertical diffusion is
  - $\frac{\partial u}{\partial t} + u \frac{\partial u}{\partial x} = -\frac{\partial \overline{u'w'}}{\partial z} \tag{3}$
- where u is flow speed and  $\overline{u'w'}$  the turbulent vertical momentum flux. In Equation (3), the
- nonlinear term  $u \frac{\partial u}{\partial x}$  represents the dynamical advection process, while the right-hand side  $-\frac{\partial \overline{u'w'}}{\partial z}$
- represents the physical forcing process, specifically parameterized turbulent vertical transport. The
- momentum flux can be expressed as:

$$\overline{u'w'} = -K\frac{\partial u}{\partial z} \tag{4}$$

where K(z) is given by:

$$K(z) = \begin{cases} 10, & z \le H \\ 0, & z > H \end{cases} \tag{5}$$

- and H = 500m denotes the prescribed boundary layer height.
- The governing equation (3) is numerically solved using the discretization strategies and physical
- configurations summarized in Table 1.

Table 1. Configuration for idealized simulation

| $\Delta x = 25$ km, $N_x = 101$ points ( $L = 2500$ km)                                  |
|------------------------------------------------------------------------------------------|
| $\Delta z = 10 \text{m}, \ N_z = 100 \ \text{levels} \ (z_{\text{top}} = 1000 \text{m})$ |
| $u_0(x,z) = 10 \text{ m s}^{-1}$ (horizontally and vertically uniform)                   |
| Periodic conditions: $u(x = 0) = u(x = L)$                                               |
|                                                                                          |
| Top: $\overline{u'w'}_{top} = 0$                                                         |
| Time-splitting with explicit advection ( $\Delta t = 300  \mathrm{s}$ ) and fully        |
|                                                                                          |

|                        | implicit diffusion.                                 |
|------------------------|-----------------------------------------------------|
| Spatial discretization | Advection: First-order upwind scheme                |
|                        | Vertical diffusion: Second-order central difference |

The surface friction configuration in Table 1 creates deliberate localized forcing, enabling controlled investigation of non-uniform surface effects on wind noise generation mechanisms in the simulated boundary layer flow.

Based on the configurations in Table 1, we conducted two contrasting idealized numerical experiments:

#### **Ideal\_Ctrl (Control Experiment):**

• Solves both horizontal advection and vertical diffusion terms at collocated grid points  $(x_i)$ 

#### **Ideal Test (Test Experiment):**

- Solves horizontal advection at  $x_i$
- Computes vertical diffusion at staggered grid points  $(x_i + \Delta x/2)$ , with surface friction velocity  $(u_*)$  prescribed at corresponding staggered positions
  - Couples solutions through linear interpolation between grids

The two experiments are designed to isolate the impact of grid staggering on noise generation under inhomogeneous surface forcing. Both experiments are simulated for 24 hours (288 steps). Figure 7 displays the spatio-temporal distribution of the simulated u-wind from both cases. The Ideal\_Ctrl simulation (Fig. 7a) exhibits localized wind reduction at the surface forcing site, with perturbations propagating downstream smoothly, demonstrating physically consistent behavior devoid of numerical noise. In contrast, the Ideal\_Test simulation (Fig. 7b) shows that additional  $2\Delta x$  grid-scale waves emerge near the local surface friction anomaly, superimposed on the expected frictional wind response. These  $2\Delta x$  waves persist throughout the integration period with temporally quasi-constant amplitude, consistent with their behavior in CMA-GFS 3.0. Figure 7 (c-d) demonstrates that the influence of surface friction is confined to the boundary layer (below the 50th model level). Comparing Fig.7 (d) with Fig. 7 (c), the staggered coupling generates  $2\Delta x$  oscillation within the boundary layer at the point of non-uniform surface friction and these spurious noise

patterns propagate to adjacent grid points, primarily near the surface. The 6-hour simulated lowest level u-wind (Fig. 7e) confirms that these  $2\Delta x$  waves represent spurious numerical oscillations induced by dispersive energy propagation, which are most pronounced adjacent to the inhomogeneous surface forcing region. The Ideal\_Test simulation results in Fig. 7 successfully reproduce both the distinctive  $2\Delta x$  noise and their characteristic spatial distribution observed in CMA-GFS 3.0.

Figure 7. Idealized experiment results (domain (grid point number): x-grid=41-59; z-grid = 1-60): (a-b)
Temporal evolution of winds at the lowest model level in (a) Ideal\_Ctrl and (b) Ideal\_Test; (c-d) Vertical

structure at 6 hour for (c) Ideal\_Ctrl and (d) Ideal\_Test; (e) 6-h wind at the lowest model level.

The idealized experiments reveal that the noise generation involves two coupled processes: (1) dispersive effects inherent to the advection-diffusion discretization (grid staggering), and (2) selective amplification and phase organization of small-scale components in surface forcing through these dispersive mechanisms. As demonstrated in Section 2.2, the stripe patterns in wind fields of CMA-GFS 3.0 predominantly occur over landmasses or near islands. This strongly indicates that wind stripe patterns in CMA-GFS 3.0 stem from inconsistent grid staggering in its dynamical-physical coupling. When combined with non-uniform surface forcing, this grid mismatch triggers dispersive wave propagation—ultimately producing the observed  $2\Delta x$  stripe patterns.

#### 3.4 Analysis of one-dimensional linear wave solution

For the one-dimensional linear damped wave equation:

$$\frac{\partial u}{\partial t} + c \frac{\partial u}{\partial x} = -\alpha u, \qquad \alpha > 0 \tag{6}$$

- where c is the phase speed,  $-\alpha u$  the damping term and  $\alpha$  the damping coefficient.
- In Eq. (6), the term  $c \frac{\partial u}{\partial x}$  represents the advection term, while the damping term  $-\alpha u$  acts as
- physical forcing. Based on the idealized experimental configuration outlined in the preceding
- section but with explicit damping time discretization, the right-hand side term of Eq. (6) can be
- discretized through two distinct numerical treatments. The first approach computes the damping
- term directly at the advection scheme's collocated grid points  $(x_i)$ , giving

$$\Delta u_i^P = -\Delta t \alpha_i u_i \tag{7}$$

- where the subscript j represents the value at grid point  $x_j$ . Equation (8) corresponds to the
- Ideal Ctrl configuration in the idealized experiment. This lead to the solution of Eq. (6) as:

$$u_i = u_i^D e^{-\alpha t} \tag{8}$$

- where  $u_j^D = u_0 e^{ik(x_j c^D t)}$  is the wave solution of the advection term at  $x_j$  after spatiotemporal
- differencing. The specific form of phase velocity  $c^D$  depends on the finite difference scheme
- employed (Durran, 2010).
- The alternative approach, analogous to the treatment in Eqs. (1) and (2), involves performing two

- averaging operations during the physics-dynamic coupling, similar to the Ideal Test configuration.
- Consequently, the damping term in Eq. (6) leads to the following change in u at grid point  $x_i$ :

$$\Delta u_j^P = -\frac{\Delta t}{2} \left( \alpha_{j-\frac{1}{2}} \frac{u_{j-1} + u_j}{2} + \alpha_{j+\frac{1}{2}} \frac{u_j + u_{j+1}}{2} \right). \tag{9}$$

- For numerically resolved scales ( $|k\Delta x| \ll 1$ ) where Taylor expansions remain valid, the variable u
- at neighboring grid points can be approximated as:

$$u_{j\pm 1} = u_j \pm \Delta x \frac{\partial u}{\partial x}\Big|_j + \frac{\Delta x^2}{2!} \frac{\partial^2 u}{\partial x^2}\Big|_j \pm \frac{\Delta x^3}{3!} \frac{\partial^3 u}{\partial x^3}\Big|_j + O(\Delta x^4), \tag{10}$$

- where the notation  $\frac{\partial u}{\partial x}\Big|_{j}$  denotes the partial derivatives of u at location  $x_{j}$  and  $\Delta x = x_{j+1} x_{j}$  is
- the Equidistant grid space. Substituting Eq. (10) into Eq. (9) and systematically neglecting fourth-
- order terms  $(O(\Delta x^4))$ , the individual wavelike solution to Eq. (6) becomes:

$$u_{j} = u_{j}^{D} e^{-\overline{\alpha}_{j} t} e^{-i\frac{k\Delta\alpha_{j}\Delta x}{4} t} e^{\frac{k^{2}\overline{\alpha}_{j}\Delta x^{2}}{4} t} e^{i\frac{k^{3}\Delta\alpha_{j}\Delta x^{3}}{24} t}, \tag{11}$$

- where  $\bar{\alpha}_j = \frac{\alpha_{j-\frac{1}{2}} + \alpha_{j+\frac{1}{2}}}{2}$ ,  $\Delta \alpha_j = \alpha_{j+\frac{1}{2}} \alpha_{j-\frac{1}{2}}$ . Compared to Eq. (8), Eq. (11) exhibits three additional
- terms in its wave solutions at  $x_i$ :
- 1) The term  $e^{-i\frac{k\Delta\alpha_j\Delta x}{4}t}$  introduces a non-dispersive phase velocity modification.
- 2) The term  $e^{\frac{k^2 \bar{a}_j \Delta x^2}{4}t}$  enhances the wave amplitude (equivalently reducing damping).
- 3) The term  $e^{i\frac{k^3\Delta\alpha_j\Delta x^3}{24}t}$  generates wavenumber-dependent phase modifications, a characteristic
- signature of dispersive wave propagation.
- The phase velocity in Eq. (11) combines both non-dispersive and dispersive modifications to the
- phase velocity of advection  $c^D$  as  $c' = c^D + \frac{\Delta \alpha_j \Delta x}{4} \frac{k^2 \Delta \alpha_j \Delta x^3}{24}$ .
- When  $\alpha$  exhibits no spatial variation (hence  $\Delta \alpha_i = 0$ ), Eq. (11) reduces to:

$$u_j = u_j^D e^{-\overline{\alpha}_j t} e^{\frac{k^2 \overline{\alpha}_j \Delta x^2}{4} t}, \tag{12}$$

- where only the amplitude-modifying term persists, producing exclusively damping reduction
- without any phase velocity modifications (neither non-dispersive shifts nor dispersive effects).
- Comparing Eqs. (8), (11) and (12) leads to the following conclusions:
- 1) The staggered-grid discretization scheme, where advection and damping terms are
- computed at offset grid points, fundamentally modifies wave solutions when compared to
- collocated approaches.

- For spatially varying damping coefficients (α), this numerical framework introduces 309 coupled amplitude-phase distortions, including wavenumber-dependent propagation speeds 310 characteristic of dispersive systems.
  - In contrast, uniform α fields restrict the influence of staggered-grid effects solely to amplitude modulation, preserving the original non-dispersive wave kinematics.
- The reformulated Eq. (9) can be expressed as: 313

$$\Delta u_j^P = -\Delta t \bar{\alpha}_j u_j - \frac{\bar{\alpha}_j \Delta t}{4} (u_{j-1} - 2u_j + u_{j+1}) - \frac{\Delta \alpha_j \Delta t}{8} (u_{j+1} - u_{j-1}). \tag{13}$$

The right-hand side of Eq. (13) contains three distinct terms. The first term is identical to Eq. (7). The second term exhibits the form of second-order horizontal diffusion - note its negative coefficient, which actually suppresses small-scale fluctuations in the damping process itself and clearly cannot generate  $2\Delta x$  oscillations. This term corresponds to the amplitude modification term in Eq. (11), reducing the damping influence. The third term represents a second-order centered difference 318 scheme. As discussed by Durran (2010), second-order centered differencing introduces dispersion in solutions to one-dimensional linear wave equations, producing upstream-propagating noise near spikes - precisely as demonstrated in our idealized experiments. Evidently, the noise source we

examine originates specifically from the third term of the right-hand side of Eq. (13).

Idealized experiments and linear wave analysis identify wind stripe noise in CMA-GFS 3.0 as requiring the coexistence of two independent factors: Physical forcing inhomogeneity (e.g., nonuniform surface friction) and physics-dynamics coupling on staggered grids. Critically, neither factor alone can generate dispersion noise—their combined presence is strictly necessary. This explains why stripes emerge preferentially over landmasses or near islands where sharp forcing gradients intersect with the model's inherent grid architecture (Fig. 2).

# 4. Confirming the origin of wind stripe patterns in CMA GFS 3.0: sensitivity experiments

The preceding analysis demonstrates that low-level wind stripe noise over terrestrial region and near islands in CMA-GFS 3.0 stems from horizontal grid mismatch in physics-dynamics coupling. To validate this mechanism, we conduct a targeted sensitivity experiment (Exp Test) by implementing conformal grid alignment for coupled processes.

In Fig. 1, the physical-point *u*-component can be expanded as  $u_p\left(x+\frac{\Delta x}{2}\right)=u(x)+O(\Delta x)$ . In Exp\_Test, we retain solely the zeroth-order approximation, enforcing  $u_p\left(x+\frac{\Delta x}{2}\right)=u(x)$ . The same treatment is applied to the v- component  $(v_p\left(y+\frac{\Delta y}{2}\right)=v(y))$ , ensuring consistent physics-dynamics coupling for both horizontal velocity components. In correspondence with Eqs. (1)-(2), we now present the modified formulations:

$$P(u_{i,j}) = P\left(u_{i-\frac{1}{2},j}\right)$$

$$P(v_{i,j}) = P\left(v_{i,j-\frac{1}{2}}\right)$$
(14)

When mapping from physics point back to the dynamics point, we have:

$$\Delta u_{i-\frac{1}{2},j}^{P} = P(u_{i,j})\Delta t$$

$$\Delta v_{i,j-\frac{1}{2}}^{P} = P(v_{i,j})\Delta t$$
(15)

This conformal alignment ensures dynamical and physical processes are computed at coincident grid points, eliminating the need for interpolation-based coupling shown in Eqs. (1)-(2). The experimental configuration replicates the Exp\_Ctrl setup described in Section 2.2, maintaining identical initial conditions and physical parameterizations.

Compared to the results from Exp\_Ctrl (Fig. 2), Exp\_Test maintains small-scale variability in terrestrial wind fields while completely eliminating  $2\Delta x$  stripe patterns (Fig. 8). The residual inhomogeneity over land, as evidenced in Fig. 8, reflects authentic physical responses to subgrid surface forcing inhomogeneity (e.g., topographic roughness, land-use variations), distinguishing it from numerical dispersion noise.

Figure 8 same as Fig. 2, but for Exp\_Test.

Spectral analysis serves as a rigorous tool for quantifying energy distribution across frequencies. To diagnose the impacts of conformal dynamics-physics coupling on wind stripe patterns we conduct one-dimensional energy spectrum analysis separately for the *u*-component (zonal direction) and *v*-component (meridional direction). We selected the East Asian domain (70-145°E, 10-65°N) as a representative non-uniform surface region and the tropical Pacific (160°E-120°W, 20°S-20°N) as a typical uniform surface region for comparative analysis. The *u*- component was processed as a one-dimensional east-west oriented sequence for Fast Fourier Transform (FFT) analysis, while the *v*- component underwent identical treatment along the north-south axis. Through spectral analysis, the power spectral density (PSD) was estimated by computing the squared modulus of the FFT coefficients (Fig. 9).

Figure 9. PSD of (a) *u*- and (b) *v*-components at 18 hour over tropical Pacific (TP: 20°S-20°N, 160°E-120°W) and East Asia (EA: 10°-65°N, 70°-145°E) domains for Exp\_Ctrl and Exp\_Test.

The key results of PSD investigation can be summarized as follows:

- 1) Over land, both u- and v- components in Exp\_Ctrl demonstrate considerably strong energy at small scales (high wavenumbers), with no energy decay observed from  $4\Delta x$  to  $2\Delta x$ . In contrast, Exp\_Test achieves significant suppression of small-scale energy, effectively mitigating high-frequency computational noise in terrestrial regions.
- 2) Over oceanic regions, the homogeneous ocean surfaces produce broadly similar spectral characteristics in both experiments. Although this general spectral agreement suggests that the horizontally homogeneous sea surface lacks the strong physical forcing inhomogeneity required to generate prominent wind stripe patterns, minor differences can still be observed at the smallest resolved scale (2Δx). While most of the central Pacific study region features homogeneous oceanic surfaces, resolvable islands (e.g., Hawaii and numerous small islands east of Australia) introduce observable heterogeneity. As demonstrated in Fig. 10 (depicting near-surface winds over the eastern Australian waters), the Exp\_Ctrl exhibits clear stripe patterns in low-level winds over some of the islands and their upwind regions, such as parts of Fiji (16-19°S, 177°-180°E) and Vanuatu (14-20°S, 166-171°E), whereas Exp\_Test shows no such artifacts. This contrast highlights the role of surface inhomogeneity in noise generation.
- 3) Terrestrial spectra exhibit distinct energy distributions between Exp\_Ctrl and Exp\_Test across scales ranging from  $2\Delta x$  to  $8\Delta x$ . This spectral difference indicates that when

inhomogeneous surface forcing combined with staggered-grid dynamic-physical coupling, dominant  $2\Delta x$ -scale perturbations emerge consistently while statistically significant oscillations develop at discrete larger scales (e.g.,  $4\Delta x$ ).

Figure 10. 18-hour forecasts of (a) u - and (b) v - component for Exp\_Ctrl, and (c) u - and (d) v - component for Exp\_Test, at the lowest model level from CMA - GFS 3.0.

Our sensitivity analysis confirms that the noise in CMA-GFS 3.0 originate specifically from the staggered-grid coupling between dynamic and physical processes over inhomogeneous surfaces. However, the noise-free Exp\_Test configuration artificially decouples the wind field from mass fields (e.g., temperature and moisture) spatially in physical processes, violating fundamental physical constraints. We emphasize that Exp\_Test serves solely as a mechanistic diagnostic to isolate the noise source rather than a viable approach for operational model improvement.

#### 5. Conclusion and discussion

The removal of fourth-order horizontal diffusion in CMA-GFS 3.0 (Shen et al., 2023), while enhancing small-scale system simulations, has inadvertently exposed underlying  $2\Delta x$  stripe patterns in low-level wind fields that were previously suppressed by the horizontal diffusion scheme. The

results of a case study demonstrate that these stripe patterns exhibits one-dimensional oriented distribution characteristics, with meridional alignment in *u*-component and zonal alignment in *v*-component. These stripes maintain continuous presence throughout the 120-hour period, with spatially predominant occurrence over land and near islands, while exhibiting diurnal amplitude modulation. We systematically elucidate the generation mechanism of these stripe patterns through a combination of numerical experiments and theoretical analysis.

Through controlled experiments with the dynamic core alone (no physics) and examination of associated surface static fields, we conclusively exclude the possibility that either the dynamic numerics or physical parameterizations independently generate the observed  $2\Delta x$  noise. This leaves physics-dynamics coupling as the sole plausible origin, consistent with our previous findings (Chen et al., 2020) that  $2\Delta x$  computational modes may emerge specifically when dynamics and physics interact through staggered-grid coupling. Two-dimensional idealized experiments successfully reproduced  $2\Delta x$  waves generated at inhomogeneous surface friction points when combining advection and friction on staggered grids. Linear wave theory analysis further revealed that this dispersion effect originates from the coupling between dynamic advection and physical forcing on staggered grids under non-uniform physical forcing. These mechanistic analyses indicate that the grid mismatch in dynamic-physical coupling is the fundamental cause of noise generation over inhomogeneous surfaces.

Sensitivity experiments confirm that when dynamic-physical grid consistency is maintained and interpolation coupling is avoided, the wind field noise can be completely eliminated. Spectral analyses reveal land-ocean contrasts: Over oceans, staggered-grid coupling shows negligible impacts due to surface homogeneity. Over land, grid mismatch in physics-dynamics coupling under inhomogeneous surface elevates small-scale energy  $(2\Delta x-8\Delta x)$ , generating broadband noise, while unstaggered physics-dynamics coupling restores the energy spectrum to a reasonable decay curve with decreasing scale.

Our results demonstrate that the stripe noise in low-level wind field of CMA-GFS 3.0 stems from grid mismatch in physics-dynamics coupling under non-uniform forcing conditions. These findings offer critical insights for other NWP models employing similar grid configurations, revealing potential numerical dispersion risks in physical-dynamics coupling on traditional staggered grids.

Building on these findings, we systematically evaluate potential solutions to mitigate wind-field noise in CMA-GFS 3.0. Since surface inhomogeneity is inherently present in NWP models, one essential approach to noise reduction involves resolving the grid mismatch between physics parameterizations and dynamical core. In latitude-longitude grid configurations, the Arakawa C-grid exhibits superior dispersion characteristics compared to unstaggered A-grid arrangements, particularly in maintaining proper phase relationships for geostrophic adjustment processes. However, adopting an A-grid configuration may introduce computational challenges in the dynamical core that could degrade numerical accuracy.

It must be emphasized that grid configuration represents a fundamental architectural feature of numerical modeling systems. As the cornerstone of model design, grid setup is determined at the earliest development stage. Redesigning grids equates to building an entirely new model system, making modifications to CMA-GFS's existing Arakawa C-grid configuration unfeasible. An alternative approach would involve computing wind fields directly on the dynamics grid within physical parameterizations. However, this strategy encounters substantial technical obstacles owing to the inherently tight coupling between wind and temperature/moisture fields in physical processes, especially when managing their interactions on staggered grids.

A compromise solution involves employing higher-order horizontal diffusion or filtering which is widely implemented in NWP models. Methods like high-order horizontal diffusion can effectively eliminate  $2\Delta x$  high-frequency noise while retaining most resolvable-scale energy. Specifically, as demonstrated by Xue (2000), 4th-order horizontal diffusion preserves about 80% of  $4\Delta x$ -scale spectral energy, and 6th-order diffusion achieves ~90% retention at this scale. However, one-dimensional linear wave analysis shows that staggered grid coupling under non-uniform forcing not only produces dispersion effects but also alters wave amplitude and phase speed, and these systematic deviations cannot be corrected by simple noise-removal tools.

Higher-order horizontal diffusion can serve as a practical remedy for unexplained noise. However, this study has definitively pinpointed the specific sources of wind-field noise in CMA-GFS. We therefore propose the following targeted recommendations:

1. While the piecewise-constant sampling method (Eqs. (15) and (16)) effectively suppresses numerical noise, it may introduce directional biases. A more physically consistent approach would be upwind sampling, where:

$$u_p\left(x + \frac{\Delta x}{2}\right) = u(x)$$
 if  $u \ge 0$  
$$u_p\left(x + \frac{\Delta x}{2}\right) = u(x + \Delta x)$$
 otherwise (16)

This straightforward approach offers immediate operational feasibility and can be rapidly implemented to address the issue. We recommend trial implementation in the operational system followed by comprehensive impact assessment.

- 2. The strong connection between wind and heat transfer in the boundary layer turbulent diffusion makes it difficult to compute momentum diffusion directly at wind grid points. As demonstrated in Chen et al. (2020), interpolating the diffusivity (rather than prognostic variables) effectively eliminated vertical grid-scale noise in thermodynamic fields a numerical artifact originally induced by staggered-grid coupling between dynamic and physical processes. Following this approach, we recommend averaging the diffusion coefficient back to the wind points and performing the vertical diffusion on the wind points, thereby avoiding interpolation of prognostic wind variables. Given its demonstrated efficacy in addressing similar discretization challenges in our previous studies (Chen et al., 2020), this approach merits implementation and systematic evaluation.
- 3. In developing next-generation model frameworks, the coordination of dynamic-physical coupling should be a core design consideration to fundamentally prevent such issues.

Data and code availability. The CMA-GFS 3.0 operational system code has been archived at <a href="https://doi.org/10.5281/zenodo.16516966">https://doi.org/10.5281/zenodo.16516966</a> (Chen, 2025). The idealized test code for diagnosing stripe patterns, along with all analysis scripts and data used to generate the figures, is available at <a href="https://doi.org/10.5281/zenodo.15597504">https://doi.org/10.5281/zenodo.15597504</a> (Chen et al., 2025).

Author contributions. JC conducted the model simulations, designed the idealized experiments, performed the diagnostic analysis, and wrote the manuscript. YS first identified the stripe patterns in operational forecasts and excluded the dynamical core's contribution through controlled numerical tests. All other authors participated in result interpretation, provided critical feedback during manuscript revisions, and contributed to figure improvements.

- Competing interests. The contact author has declared that none of the authors has any competing
- interests.

- Acknowledgements. This research was supported by the NSFC Major Project (42090032) and
- Beijing Natural Science Foundation for Young Scholars (8254056).

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
