# Peer review of "Stripe Patterns in Wind Forecasts Induced by Physics-Dynamics"

_EGUsphere, 2025_

## Referee Comment (RC1)

**Stripe Patterns in Wind Forecasts Induced by Physics-Dynamics Coupling on a Staggered Grid in CMA-GFS 3.0 by Jiong Chen, Yong Su, Zhe Li, Zhanshan Ma and Xueshun Shen**

This is a very interesting investigation into the cause of, and aspects of a remedy for, the appearance of stripes in wind forecasts. The authors:

- present the problem clearly,
- discuss the possible cause,
- present the results of an idealised model to support their hypothesis,
- develop a theoretical model to show what is going on,
- and then show the positive impact of one approach to avoiding the problem.

This is a great example of the scientific method at work and the results are all clearly presented.

I have only one comment that I have listed as a main point which is a suggestion for how I think the authors could give some more insight into what is going on and that would add some value to the presentation.

Other than that, I am happy to recommend publication subject to various rather minor comments/questions/suggestions listed below. The length of some of these might suggest a more major nature but my intention is that of Minor Revision.

**Main point**

The one aspect of the presentation that I think could be improved is for the authors to give the reader better clarity on what is going on and how it leads to the results seen.

In terms of the analysis section 3.4 there are two models of the damping process:

$$\Delta u^\alpha_{i+\frac{1}{2}} = -\Delta t \, \alpha_{i+\frac{1}{2}} \, u_{i+\frac{1}{2}}, \quad \dots (A)$$

and

$$\Delta u^\alpha_{i+\frac{1}{2}} = -\frac{\Delta t}{2}\left[\left(\overline{\alpha}_{i+\frac{1}{2}} + \Delta\alpha_{i+\frac{1}{2}}\right)\frac{1}{2}\left(u_{i+\frac{3}{2}} + u_{i+\frac{1}{2}}\right) + \left(\overline{\alpha}_{i+\frac{1}{2}} - \Delta\alpha_{i+\frac{1}{2}}\right)\frac{1}{2}\left(u_{i+\frac{1}{2}} + u_{i-\frac{1}{2}}\right)\right]. \quad \dots (B)$$

There are two steps that lead to noise being induced by process B.

The first step is common to both A and B. It is perhaps clear to many readers but I think it would be worth being explicit that applying either A or B to even (or perhaps especially) an initially constant field will induce a variation in that field that reflects the variation of $\alpha$. Such an effect is clear from the results of the control experiment in Fig 7e where the initially uniform wind is damped to zero at one point but remains almost at its initial value upstream of that point – the horizontal variation of $u_*$ is reflected in the wind field.

The next step in the argument is that for process B, the combination of horizontal variation in $\alpha$ and averaging of fields back and forth leads to a dispersive error in how the wind field evolves

that is not seen for process A (for which the impact of the damping coefficient is purely local). That this is the case can be seen by rewriting process B as:

$$\Delta u^{\alpha}_{i+\frac{1}{2}} = -\frac{\overline{\alpha}_{i+\frac{1}{2}}\Delta t}{4}\left(u_{i+\frac{3}{2}} + 2u_{i+\frac{1}{2}} + u_{i-\frac{1}{2}}\right) - \frac{\Delta\alpha_{i+\frac{1}{2}}\Delta t}{4}\left(u_{i+\frac{3}{2}} - u_{i-\frac{1}{2}}\right). \quad \dots (C)$$

In this form two things become apparent:

1. The first term is perhaps not surprising. It has the form of a second-order accurate horizontal diffusion scheme. Any $2\Delta x$ mode, that emerges by whatever route, is invisible to this term and so will not be damped at all. Other modes will also be impacted by the smoothing inherent in the 1-2-1 operator. (In contrast, process A will damp all modes equally as efficiently.)
2. More interestingly perhaps, the second term can be interpreted as a second-order centred-difference advection scheme where the advecting "velocity" is $\Delta x\Delta\alpha/2$. This term is therefore not damping at all. It is also the source of the dispersion issue: When applied to a field with a discontinuous field such a scheme is well known to create upstream propagating noise. Indeed, Fig 3.7a of Durran's second edition of Numerical Methods for Fluid Dynamics is very reminiscent of the form of the result of process B shown in Fig 7e of the present work.

None of this says a lot more than the authors already present in their work but I feel that adding a brief discussion of the form (C) and making the analogy with centred advection (and its dispersion error and associated noise) might, for some readers, give a bit more insight into what is happening.

**Minor comments**

(Lnn refers to line number nn.)

1. L13: I am not convinced that the comment about the absence of noise in the static fields is quite correct – see my comment below about L177. I would suggest instead saying something like 'the structure of the static fields in not consistent with the amplitude of the $2\Delta x$ noise if that noise were forced locally by the static fields'?
2. L80-83: The UK Met Office model does *not* follow this approach. It averages the winds to the cell centre but only uses these averaged winds to evaluate the boundary layer diffusion coefficient (the eddy diffusivity). It then averages the diffusion coefficient back to the wind points and performs the vertical diffusion at the wind points. This is the approach that I would recommend in solving the problem presented in this paper if your infrastructure can support such an approach.
3. L156-157: It would be good somewhere to comment on the stability of the model, i.e. is the amplitude of the stripes approximately constant in that they appear and remain approximately unchanged, or do they grow in time?
4. L177: Without further evidence to support this, I think 'complete absence' is too strong. It is clear that there is not the same visual level of noise at the $2\Delta x$ scale as in Fig 2 but that is not the same as there being a complete absence. And given that horizontal averaging of a field cannot create a $2\Delta x$ component then I think that the heart of the later argument lies in there being some forcing of such a component by the physics. It would be interesting to present a spectral analysis of these fields in the same way as Fig 9.

5. L182: It would be useful to the reader to give an indicative latitude and longitude for where the islands are.
6. L209, section 3.3: I think here one is looking for the simplest experimental set-up that shows the noisy behaviour. Given the theoretical model used in section 3.4 I would have thought that it would be best in this section to match as closely as possible that theoretical model, i.e. use a constant advecting wind (thereby losing the complication of nonlinearity) and use a constant eddy viscosity in the vertical (thereby losing the complication of nonzero values of $dK/dz$ . If the hypothesis is correct then the equivalent figures to Figs 7a, b and e should be very similar.
7. L216: If the authors do retain a height varying $K$ then it is probably worth saying that this is an example profile for the purposes of the idealised model rather than the profile used in the full model.
8. Equation (5): I believe there is a missing factor of $u_*$.
9. L220/236: To save the reader having to work it out for themselves, it would be good to state that the time step is 300s (if I have worked it out correctly!).
10. L241: It would be useful to say whether the noise grows or is stable.
11. L252, Fig 7: It would be worth stating in the caption that 'x-grid' means the grid point number not a distance.
12. L255-256: I am not convinced that the words used for either source of 'noise' are correct. Earlier it has been stated that there is a complete absence of noise in the surface fields of the full model! The surface forcing is what it is – it is the amplification, or exposure (through dispersion), of whatever $2\Delta x$ component that is at question. Also, I think it is debatable whether that averaging is an inconsistency; I would suggest that it is a discretization choice (albeit perhaps not a good choice!).
13. L272 and following lines: The authors have included one part of the averaging process, from the physics points (cell centres) to the velocity points (cell faces). They have then used Taylor series expansions to estimate the winds at the cell centres. It would be better (and no more difficult) to explicitly include the averaging of the winds to the cell centres and then use Taylor series expansions to obtain estimates for $u(x \pm \Delta x)$ in terms of $u(x)$. This has the effect of changing, in Eq. (10), the denominator 8 to 4, and the denominator 48 to 12. Although this is only a minor change in practice, it better reflects what the model of section 3.3 actually does, and perhaps more usefully, the terms that are proportional to $\Delta\alpha$ are proportional to $u' + \Delta x^2 u'''/6$ which are the terms that arise in a second-order centred advection scheme (as expected from the analysis suggested in the section Main Point).
14. L275: Strictly, Eq. (10) is only the solution to Eq. (8) if $\overline{\alpha}$ and $\Delta\alpha$ are taken as independent of $x$ which they cannot both be over a periodic domain. You could either rewrite (8) in terms of $\overline{\alpha}$ and $\Delta\alpha$ and then postulate a different problem that has those coefficients constant, or simply say that the solution is approximate and only holds locally.
15. L277: Related to the above point, I think here (and everywhere they are used) $\overline{\alpha}$ and $\Delta\alpha$ should be written as $\overline{\alpha}_{i+1/2}$ and $\Delta\alpha_{i+1/2}$ to make their horizontal dependency clear.
16. L277: It would be much better to not include the factor of 1/2 in the definition of $\Delta\alpha$ but carry that explicitly where $\Delta\alpha$ is used. Otherwise the definition is not consistent with the use of $\Delta$ elsewhere, e.g., consider what happens if $\alpha \equiv x$, we would then end up with $\Delta x = \Delta x/2$!
17. L280: It would be worth reminding the reader somewhere that the Taylor series expansion is only valid for small values of $k\Delta x$. Otherwise, when $k\Delta x = \pi$, as it can do,

the solution (both (10) and (11)) would be predicted to grow unlimitedly since $\pi^2/8 - 1 > 0$.

18. L303: The approach of this section makes sense and is a nice way of showing that the noise can be controlled. However, it is perhaps worth pointing out that the approach used to do so introduces an arbitrary bias in the direction in which the now piecewise-constant sampling is done, i.e. why choose $u(x)$ instead of $u(x + \Delta x)$? I appreciate that this would not be practicable as a quick experiment, but a better approach might be to always sample upwind, e.g., $u_p(x + \Delta x/2) = u(x)$ when $u > 0$ and $= u(x + \Delta x)$ otherwise.
19. L310: Please say whether this is also applied for the reverse mapping from the physics point back to the dynamics point.
20. L342: I feel that there is still an interesting level of difference in the spectra over the sea, more than 'remarkable consistency' would suggest.
21. L401-409: It is a shame if it is not possible within CMA-GFS to apply the boundary-layer code to wind fields on their native points. This would seem to me to be the preferred solution rather than reintroducing horizontal diffusion. And, as I noted earlier, this is the approach taken within the Met Office's Unified Model.

**Typos/editorial comments**

1. L13: 'dynamical core' is more standard than 'dynamic-core'
2. L21: 'physics-dynamics' is more standard than 'physics-dynamic'
3. L130: word missing from 'therefore systematically these'
4. L243: 'Compared' should be 'Comparing'
5. L280: There is an overbar missing from the $\alpha$
6. L284: The exponent in the middle term, together with the factor $i$, need to be removed

Nigel Wood

July 2025

---

## Referee Comment (RC3)

**Review of Stripe Patterns in Wind Forecasts Induced by Physics-Dynamics Coupling on a Staggered Grid in CMA-GFS 3.0 by Chen et. al.**

Sean Patrick Santos

September 26, 2025

This manuscript is an intriguing analysis of how physics-dynamics coupling can lead to numerical artifacts (striping) when the physics and dynamical processes are evaluated using data non-collocated grid points. It combines, in my view successfully, results from global model experiments, a simplified numerical model demonstrating similar difficulties, and a theoretical analysis of how staggered winds may lead to dispersive behavior. The presentation of most points is clear and convincing, and most of my concerns with the originally submitted draft have already been corrected in the response to Nigel Wood.

I have two minor concerns outstanding. First, the equation (7) in the manuscript is not the analytic solution to (6) if $\alpha$ is spatially varying. To give an example, if $\alpha = \alpha_0 \cos(mx)$, i.e.

$$\frac{\partial u}{\partial t} + c\frac{\partial u}{\partial x} = \alpha_0 \cos^2(mx)u \tag{1}$$

then one solution is given by:

$$u = u_0 \exp\left\{ ik(x - ct) - \frac{\alpha_0 t}{2} - \frac{\alpha_0 \sin(2mx)}{4cm} \right\} \tag{2}$$

When the length scale across which $\alpha$ varies is

---

## Author Comment (AC4)

**Response to Reviewer Comments for MS egusphere-2025-2704**

**Dear Dr. Wood,**

We sincerely appreciate your thorough evaluation and constructive suggestions for our manuscript "Stripe Patterns in Wind Forecasts Induced by Physics-Dynamic Coupling on a Staggered Grid in CMA-GFS 3.0". Your insightful comments not only affirm the value of this work but have also significantly helped us refine both the technical rigor and clarity of presentation. In particular, your detailed description of the UK Met Office model's solution to this issue has provided invaluable inspiration, guiding our next steps in adapting and implementing similar improvements in CMA-GFS.

Below we respond point-by-point to your comments, outlining how we plan to address them in the revised manuscript pending all reviewer feedback.

**Main point**

The one aspect of the presentation that I think could be improved is for the authors to give the reader better clarity on what is going on and how it leads to the results seen.

In terms of the analysis section 3.4 there are two models of the damping process:

$$\Delta u^\alpha_{i+\frac{1}{2}} = -\Delta t\, \alpha_{i+\frac{1}{2}}\, u_{i+\frac{1}{2}}, \quad \dots (A)$$

and

$$\Delta u^\alpha_{i+\frac{1}{2}} = -\frac{\Delta t}{2}\left[\left(\overline{\alpha}_{i+\frac{1}{2}} + \Delta\alpha_{i+\frac{1}{2}}\right)\frac{1}{2}\left(u_{i+\frac{3}{2}} + u_{i+\frac{1}{2}}\right) + \left(\overline{\alpha}_{i+\frac{1}{2}} - \Delta\alpha_{i+\frac{1}{2}}\right)\frac{1}{2}\left(u_{i+\frac{1}{2}} + u_{i-\frac{1}{2}}\right)\right]. \quad \dots (B)$$

There are two steps that lead to noise being induced by process B.

The first step is common to both A and B. It is perhaps clear to many readers but I think it would be worth being explicit that applying either A or B to even (or perhaps especially) an initially constant field will induce a variation in that field that reflects the variation of $\alpha$. Such an effect is clear from the results of the control experiment in Fig 7e where the initially uniform wind is damped to zero at one point but remains almost at its initial value upstream of that point – the horizontal variation of $u_*$ is reflected in the wind field.

The next step in the argument is that for process B, the combination of horizontal variation in $\alpha$ and averaging of fields back and forth leads to a dispersive error in how the wind field evolves that is not seen for process A (for which the impact of the damping coefficient is purely local). That this is the case can be seen by rewriting process B as:

$$\Delta u^\alpha_{i+\frac{1}{2}} = -\frac{\overline{\alpha}_{i+\frac{1}{2}}\Delta t}{4}\left(u_{i+\frac{3}{2}} + 2u_{i+\frac{1}{2}} + u_{i-\frac{1}{2}}\right) - \frac{\Delta\alpha_{i+\frac{1}{2}}\Delta t}{4}\left(u_{i+\frac{3}{2}} - u_{i-\frac{1}{2}}\right). \quad \dots (C)$$

In this form two things become apparent:

1. The first term is perhaps not surprising. It has the form of a second-order accurate horizontal diffusion scheme. Any $2\Delta x$ mode, that emerges by whatever route, is invisible to this term and so will not be damped at all. Other modes will also be impacted by the smoothing inherent in the 1-2-1 operator. (In contrast, process A will damp all modes equally as efficiently.)

2. More interestingly perhaps, the second term can be interpreted as a second-order centred-difference advection scheme where the advecting "velocity" is $\Delta x \Delta \alpha / 2$. This term is therefore not damping at all. It is also the source of the dispersion issue: When applied to a field with a discontinuous field such a scheme is well known to create upstream propagating noise. Indeed, Fig 3.7a of Durran's second edition of Numerical Methods for Fluid Dynamics is very reminiscent of the form of the result of process B shown in Fig 7e of the present work.

None of this says a lot more than the authors already present in their work but I feel that adding a brief discussion of the form (C) and making the analogy with centred advection (and its dispersion error and associated noise) might, for some readers, give a bit more insight into what is happening.

**Your comments are highly insightful and have prompted us to more deeply consider the origin of the dispersion error demonstrated in our study. Eq. (C) is particularly interesting as it further clarifies that the true dispersion error stems from the second term in the equation. In response to your suggestions, we propose the following modifications:**

1. **The variables configuration in staggered coupling using wave analysis is demonstrated in Figure R1. This configuration maintains consistent subscript notation with the formulas in Durran's book (Section 3.3).**

[Figure]

**Figure R1. The configuration of second model of damping process shown as Eq. (10) in the following revised version**

2. **This study focuses solely on the numerical coupling impact of local friction terms, not advection discretization—a distinction clarified in the revised manuscript. Our idealized experiments demonstrate that spatially varying friction induces non-uniform $u$ distributions (e.g., spiked structures), whose differential responses depend on the discretization scheme. As Durran's description in Section 3.3 of his book, central differencing of advection introduces dispersion artifacts near discontinuities, which is also why we employ first-order upwind scheme for advection in our idealized experiments to isolate the physics-dynamic coupling issue in this study.**

3. **As noted in your Comment #13, our analysis in the manuscript considered only one component of the averaging process—specifically, the physics-to-dynamics interpolation (Eq.2) without accounting for the dynamics-to-physics averaging (Eq. 1). Following your**

**suggestion, we confirm this revised derivation does yield additional terms with modified coefficients, as you anticipated (See Eq. (12), here** $\Delta\alpha_j = \alpha_{j+\frac{1}{2}} - \alpha_{j-\frac{1}{2}}$ **based on your comment #16).**

4. **Following your suggestion, we will incorporate a discussion of Equation C, which provides deeper insight into the underlying mechanism—benefiting both readers and our own understanding of this phenomenon's root cause.**

**To reflect these, we will modify Line 263-296 as follows:**

For the one-dimensional linear damped wave equation:

$$\frac{\partial u}{\partial t} + c\frac{\partial u}{\partial x} = -\alpha u, \qquad \alpha > 0, \tag{6}$$

where $c$ is the phase speed, $-\alpha u$ the damping term and $\alpha$ the damping coefficient. If the $x$-domain is periodic, the analytic solution for each individual mode takes the form:

$$u = u_0 e^{-\alpha t} e^{ik(x-ct)}. \tag{7}$$

Here $u_0$ is the initial amplitude and $k$ is the wavenumber. Equation (7) describes an exact, non-dispersive wave propagation where all wavenumbers propagate at identical phase speed $c$.

In Eq. (6), the term $c\frac{\partial u}{\partial x}$ represents the advection term, while the damping term $-\alpha u$ acts as physical forcing. Based on the idealized experimental configuration outlined in the preceding section but with explicit damping time discretization, the right-hand side term of Eq. (6) can be discretized through two distinct numerical treatments. The first approach computes the damping term directly at the advection scheme's collocated grid points ($x_j$), giving

$$\Delta u_j^P = -\Delta t \alpha_j u_j, \tag{8}$$

where the subscript $j$ represents the value at grid point $x_j$. Equation (8) corresponds to the Ideal_Ctrl configuration in the idealized experiment. This lead to the solution of Eq. (6) as:

$$u_j = u_j^D e^{-\alpha t}, \tag{9}$$

where $u_j^D$ is the wave solution of the advection term at $x_j$ after spatiotemporal differencing. The specific form of $u_j^D$ depends on the finite difference scheme employed (Durran, 2010). In Eq. (9), the numerical solution of the discretized damping term matches the analytical solution in Eq. (7), where the damping term solely causes exponential decay of the wave amplitude.

The alternative approach, analogous to the treatment in Eqs. (1) and (2), involves performing two averaging operations during the physics-dynamic coupling, similar to the Ideal_Test configuration.

Consequently, the damping term in Eq. (6) leads to the following change in $u$ at grid point $x_j$:

$$\Delta u_j^P = -\frac{\Delta t}{2}\left(\alpha_{j-\frac{1}{2}}\frac{u_{j-1}+u_j}{2}+\alpha_{j+\frac{1}{2}}\frac{u_j+u_{j+1}}{2}\right). \tag{10}$$

For numerically resolved scales ($|k\Delta x|\leq\pi/2$) where Taylor expansions remain valid, the variable $u$ at neighboring grid points can be approximated as:

$$u_{j\pm1}=u_j\pm\Delta x\left.\frac{\partial u}{\partial x}\right|_j+\frac{\Delta x^2}{2!}\left.\frac{\partial^2 u}{\partial x^2}\right|_j\pm\frac{\Delta x^3}{3!}\left.\frac{\partial^3 u}{\partial x^3}\right|_j+O(\Delta x^4), \tag{11}$$

where the notation $\left.\frac{\partial u}{\partial x}\right|_j$ denotes the partial derivatives of $u$ at location $x_j$ and $\Delta x = x_{j+1}-x_j$ is the Equidistant grid space. Substituting Eq. (11) into Eq. (10) and systematically neglecting fourth-order terms ($O(\Delta x^4)$), the individual wavelike solution to Eq. (6) becomes:

$$u_j=u_j^D e^{-\bar{\alpha}_j t}e^{-i\frac{k\Delta\alpha_j\Delta x}{4}t}e^{\frac{k^2\bar{\alpha}_j\Delta x^2}{4}t}e^{i\frac{k^3\Delta\alpha_j\Delta x^3}{24}t}, \tag{12}$$

where $\bar{\alpha}_j=\frac{\alpha_{j-\frac{1}{2}}+\alpha_{j+\frac{1}{2}}}{2}, \Delta\alpha_j=\alpha_{j+\frac{1}{2}}-\alpha_{j-\frac{1}{2}}$. Equation (12) exhibits three additional terms in its wave solutions at $x_j$ compare to Eq. (9):

1) The term $e^{-i\frac{k\Delta\alpha_j\Delta x}{4}t}$ introduces a non-dispersive phase velocity modification.

2) The term $e^{\frac{k^2\bar{\alpha}_j\Delta x^2}{4}t}$ enhances the wave amplitude (equivalently reducing damping).

3) The term $e^{i\frac{k^3\Delta\alpha_j\Delta x^3}{24}t}$ generates wavenumber-dependent phase modifications, a characteristic signature of dispersive wave propagation.

The phase velocity in Eq. (12) combines non-dispersive and a dispersive modifications as $c'=c^D+\frac{k\Delta\alpha_j\Delta x}{4}+\frac{k^3\Delta\alpha_j\Delta x^3}{24}$, where $c^D$ is the phase velocity of advection, depending on the numeric scheme applied.

When $\alpha$ exhibits no spatial variation (hence $\Delta\alpha_j=0$), Eq. (12) reduces to:

$$u_j=u_j^D e^{-\bar{\alpha}_j t}e^{\frac{k^2\bar{\alpha}_j\Delta x^2}{4}t}, \tag{1}$$

where only the amplitude-modifying term persists, producing exclusively damping reduction without any phase velocity modifications (neither non-dispersive shifts nor dispersive effects).

Comparing Eqs. (9), (12) and (13) leads to the following conclusions:

1) The staggered-grid discretization scheme, where advection and damping terms are computed at offset grid points, fundamentally modifies wave solutions when compared to collocated approaches.

2) For spatially varying damping coefficients (α), this numerical framework introduces coupled amplitude-phase distortions, including wavenumber-dependent propagation speeds characteristic of dispersive systems.

3) In contrast, uniform α fields restrict the influence of staggered-grid effects solely to amplitude modulation, preserving the original non-dispersive wave kinematics.

**Then we will add the discussion of Eq.(C). In this part, we modified Eq.(C) into three terms.**

The reformulated Eq. (10) can be expressed as:

$$\Delta u_j^P = -\Delta t \bar{\alpha}_j u_j - \frac{\bar{\alpha}_j \Delta t}{4}\left(u_{j-1} - 2u_j + u_{j+1}\right) - \frac{\Delta \alpha_j \Delta t}{8}\left(u_{j+1} - u_{j-1}\right). \tag{14}$$

The right-hand side of Eq. (14) contains three distinct terms. The first term is identical to Eq. (8). The second term exhibits the form of second-order horizontal diffusion - note its negative coefficient, which actually suppresses small-scale fluctuations in the damping process itself and clearly cannot generate $2\Delta x$ oscillations. This term corresponds to the amplitude modification term in Eq. (12), reducing the damping influence. The third term represents a second-order centered difference scheme. As discussed by Durran (2010), second-order centered differencing introduces dispersion in solutions to one-dimensional linear wave equations, producing upstream-propagating noise near spikes - precisely as demonstrated in our idealized experiments. Evidently, the noise source we examine originates specifically from the third term of the right-hand side of Eq. (14).

**Line 297-302 will be unchanged.**

**Minor comments**

(Lnn refers to line number nn.)

1. L13: I am not convinced that the comment about the absence of noise in the static fields is quite correct – see my comment below about L177. I would suggest instead saying something like 'the structure of the static fields in not consistent with the amplitude of the $2\Delta x$ noise if that noise were forced locally by the static fields'?

**We appreciate your suggestion and will adopt the following more rigorous description in our revised manuscript:**

The structural mismatch between static field variations and the observed $2\Delta x$ noise amplitude suggests that locally forced mechanisms from surface inhomogeneity alone cannot explain the wind stripe patterns. Meanwhile, pure dynamical core simulations exhibit no such noise, confirming that

the dynamical core itself does not generate these patterns.

> 2. L80-83: The UK Met Office model does *not* follow this approach. It averages the winds to the cell centre but only uses these averaged winds to evaluate the boundary layer diffusion coefficient (the eddy diffusivity). It then averages the diffusion coefficient back to the wind points and performs the vertical diffusion at the wind points. This is the approach that I would recommend in solving the problem presented in this paper if your infrastructure can support such an approach.

**We sincerely appreciate you sharing how the UK Met Office model addresses this issue. Based on the analysis presented in this study, we are confident that this approach will effectively resolve the noise problem. We will implement and evaluate corresponding improvements in CMA-GFS. Please also refer to our response to Comment #21 for additional clarification on this point.**

> 3. L156-157: It would be good somewhere to comment on the stability of the model, i.e. is the amplitude of the stripes approximately constant in that they appear and remain approximately unchanged, or do they grow in time?

**As shown in Figure 3, the amplitude of the stripes remains approximately unchanged with forecast days despite pronounced diurnal variations. Therefore, we will modify the original sentence to:**

The $2\Delta x$ oscillations exhibit continuous presence throughout the integration period. Their amplitude shows strong diurnal variations but no systematic growth with forecast days, suggesting that this noise is unlikely to directly induce model instability.

> 4. L177: Without further evidence to support this, I think 'complete absence' is too strong. It is clear that there is not the same visual level of noise at the $2\Delta x$ scale as in Fig 2 but that is not the same as there being a complete absence. And given that horizontal averaging of a field cannot create a $2\Delta x$ component then I think that the heart of the later argument lies in there being some forcing of such a component by the physics. It would be interesting to present a spectral analysis of these fields in the same way as Fig 9.

**We sincerely appreciate your attention to this detail. Spectral analysis in Fig. R2 demonstrates small-scale fluctuations in both standard deviation of sub-grid orography and roughness length, which are also visually evident in Fig. R3. These results demonstrate the existence of small-scale fluctuations in the static fields. However, as shown in Fig. 5 of our manuscript, these static fields do not exhibit the stripe patterns observed in the wind field distribution, thereby excluding them as direct sources of the stripe noise. Their spectral characteristics differ from the wind field noise: the static fields' energy spectra decay at smaller scales (Fig. R2), whereas the Ctrl_EA wind field spectra intensify at these scales (Fig. 9). This contrast suggests that small-scale static inhomogeneities are probably not the primary direct source of the observed wind stripes.**

**Critically, static field inhomogeneity (particularly the prominent spikes in Fig. R3) serves as triggering factors for the noise through wave dispersion effects - as discussed in Sections 3.3 and 3.4. Based on these considerations, the term 'complete absence' is scientifically inappropriate.**

**To align with the overall logic of our analysis, we will modify Lines 177–179 as follows:**

The distributions of the two static fields in Fig. 5 differ markedly from the wind stripe patterns in Fig. 2, indicating that surface parameter inhomogeneity is not the direct origin of the stripe noise.

[Figure]

**Figure R2. PSD of static field over the East Asia (70°-145°E, 10°-65°N): roughness length along $x$-direction: solid line; roughness length along $y$-direction: dashed line; orography standard deviation along $x$-direction: dash-dot line; orography standard deviation along $y$-direction dotted line**

[Figure]

**Figure R3. The roughness length along 100°E**

5. L182: It would be useful to the reader to give an indicative latitude and longitude for where the islands are.

**We appreciate this suggestion. The islands discussed in our study are located at approximately 93°E, 11–13°N. This information will be added to the revised manuscript.**

The Andaman Islands (11–13°N, 93°E),

6. L209, section 3.3: I think here one is looking for the simplest experimental set-up that shows the noisy behaviour. Given the theoretical model used in section 3.4 I would have thought that it would be best in this section to match as closely as possible that theoretical model, i.e. use a constant advecting wind (thereby losing the complication of nonlinearity) and use a constant eddy viscosity in the vertical (thereby losing the complication of nonzero values of $dK/dz$. If the hypothesis is correct then the equivalent figures to Figs 7a, b and e should be very similar.

We sincerely appreciate this insightful suggestion. As you correctly noted, both linear advection and constant eddy viscosity ($K$) successfully reproduce the $2\Delta x$ waves near surface friction point (Fig. R4). However, our analysis reveals that nonlinear advection amplifies these $2\Delta x$ fluctuations, which may explain why the stripe noise is so obvious in the CMA-GFS's wind forecasts. Additionally, larger $K$ values enhance the friction impact on upper boundary layer (Fig. R4, Right panel).

To balance realism with simplicity, we retain nonlinear advection (to best replicate the full model's behavior) while adopting constant $K$ for idealized experiments. This configuration (Fig. R4, middle panel) will be selected for consistency, and we will accordingly revise Line 215 and Equation (5) as

where K is given by:

$$\begin{cases} K(z) = 10, & z \leq H \\ 0, & z > H \end{cases} \tag{2}$$

[Figure]

Figure R4. Left: Linear advection with $c$=10m s$^{-1}$, but retain height-dependent $K$; Middle: nonlinear advection with constant $K$=10 m$^2$ s$^{-1}$; Right: nonlinear advection with constant $K$=50 m$^2$ s$^{-1}$.

7. L216: If the authors do retain a height varying $K$ then it is probably worth saying that this is an example profile for the purposes of the idealised model rather than the profile used in the full model.

Thank you for your suggestion. Consistent with our stated experimental design (Response #6), we will employ the constant eddy viscosity ($K$) configuration for the idealized simulations.

8. Equation (5): I believe there is a missing factor of $u_*$.

We thank you for prompting deeper consideration of this issue. As shown in Fig. R5, $u_*$-dependent $K$ becomes highly localized (peaking at the friction point). With a constant 500-m boundary layer height, our setup approximates an unstable boundary layer in all $x$ points. Since unstable boundary layer involves both thermal-driven turbulence effects (absent here) and wind shear-driven turbulence, using $u_*$-dependent $K$ would demand simultaneous heat flux modeling. Our constant-$K$ choice (as detailed in Response #6) avoids this complexity while maintaining physical consistency with the prescribed constant boundary layer depth.

[Figure]

**Figure R5. Linear advection with $u_*$-dependent $K$**

9. L220/236: To save the reader having to work it out for themselves, it would be good to state that the time step is 300s (if I have worked it out correctly!).

**I am very sorry I forgot to state it. The time step is 300s and it will be added in Table 1. Thank you very much for your careful attention to this detail.**

10. L241: It would be useful to say whether the noise grows or is stable.

**Thank you for this important clarification. As shown in Fig. 7b, the $2\Delta x$ wave amplitude remains nearly constant throughout the integration period, mirroring the behavior in CMA-GFS. This stability will be explicitly stated in the revised manuscript:**

These $2\Delta x$ waves persist throughout the integration period with temporally quasi-constant amplitude, consistent with their behavior in CMA-GFS 3.0.

11. L252, Fig 7: It would be worth stating in the caption that 'x-grid' means the grid point number not a distance.

**We appreciate this clarification. The caption will be revised as:**

Figure 7. Idealized experiment results (domain (grid point number): x-grid = 41-59; z-grid = 1-60):

> 12. L255-256: I am not convinced that the words used for either source of 'noise' are correct. Earlier it has been stated that there is a complete absence of noise in the surface fields of the full model! The surface forcing is what it is – it is the amplification, or exposure (through dispersion), of whatever $2\Delta x$ component that is at question. Also, I think it is debatable whether that averaging is an inconsistency; I would suggest that it is a discretization choice (albeit perhaps not a good choice!).

**We appreciate these nuanced observations. The wording will be to more precisely characterize the mechanisms:**

1. **Surface effects: The original text suggested surface forcing directly creates noise, which was inaccurate. Instead, surface roughness variations (Fig. R3) interact with the model's numerical dispersion, amplifying pre-existing 2Δx waves.**
2. **Grid staggering: The term 'inconsistent' was misleading. The issue arises because advection and diffusion use different grid arrangements (staggered vs. non-staggered), which can artificially enhance certain wave modes.**

**This sentence will be modified as:**

The idealized experiments reveal that the noise generation involves two coupled processes: (1) dispersive effects inherent to the advection-diffusion discretization (grid staggering), and (2) selective amplification and phase organization of small-scale components in surface forcing through these dispersive mechanisms.

> 13. L272 and following lines: The authors have included one part of the averaging process, from the physics points (cell centres) to the velocity points (cell faces). They have then used Taylor series expansions to estimate the winds at the cell centres. It would be better (and no more difficult) to explicitly include the averaging of the winds to the cell centres and then use Taylor series expansions to obtain estimates for $u(x \pm \Delta x)$ in terms of $u(x)$. This has the effect of changing, in Eq. (10), the denominator 8 to 4, and the denominator 48 to 12. Although this is only a minor change in practice, it better reflects what the model of section 3.3 actually does, and perhaps more usefully, the terms that are proportional to $\Delta\alpha$ are proportional to $u' + \Delta x^2 u'''/6$ which are the terms that arise in a second-order centred advection scheme (as expected from the analysis suggested in the section Main Point).

**We appreciate your insightful suggestion regarding the averaging process. As noted in our response to the Main Point comment, we have revised the derivation in Section 3.3 to explicitly include the averaging of winds to cell centers before applying Taylor series expansions. Following your comment #16, this adjustment now modifies the coefficients in Eq. (12) (originally Eq.(10)) to 4, 4, and 24, which more accurately represents the model's numerical formulation.**

> 14. L275: Strictly, Eq. (10) is only the solution to Eq. (8) if $\overline{\alpha}$ and $\Delta\alpha$ are taken as independent of $x$ which they cannot both be over a periodic domain. You could either rewrite (8) in terms of $\overline{\alpha}$ and $\Delta\alpha$ and then postulate a different problem that has those coefficients constant, or simply say that the solution is approximate and only holds locally.

**We agree with your observation. In Eq. (12) [modification of (10)], $\overline{\alpha}_j$ and $\Delta\alpha_j$ are**

evaluated at a given grid point $x_j$. This approach is valid for diagnosing local dispersion properties, as reflected in our response to the 'Main Point.'

> 15. L277: Related to the above point, I think here (and everywhere they are used) $\overline{\alpha}$ and $\Delta\alpha$ should be written as $\overline{\alpha}_{i+1/2}$ and $\Delta\alpha_{i+1/2}$ to make their horizontal dependency clear.

**We fully agree with this suggestion and will implement the proposed changes. Please see our response to the 'Main Point' comment.**

> 16. L277: It would be much better to not include the factor of 1/2 in the definition of $\Delta\alpha$ but carry that explicitly where $\Delta\alpha$ is used. Otherwise the definition is not consistent with the use of $\Delta$ elsewhere, e.g., consider what happens if $\alpha \equiv x$, we would then end up with $\Delta x = \Delta x/2$!

**We agree and have removed the factor of 1/2 from the definition of $\Delta\alpha_j$, carrying it explicitly in subsequent equations. This correction aligns with standard notation and avoids scaling inconsistencies. The coefficients in Eq. (12) have been updated accordingly, as detailed in our response to the 'Main Point' section.**

> 17. L280: It would be worth reminding the reader somewhere that the Taylor series expansion is only valid for small values of $k\Delta x$. Otherwise, when $k\Delta x = \pi$, as it can do, the solution (both (10) and (11)) would be predicted to grow unlimitedly since $\pi^2/8 - 1 > 0$.

**Thank you for this critical insight. As now clarified before Eq. (11), our Taylor expansions are strictly valid for numerically resolved scales ($|k\Delta x| \leq \pi/2$). While higher-order terms become significant near $k\Delta x = \pi$, this doesn't affect our analysis of $2\Delta x$ waves ($k\Delta x = \pi/2$). Line 272 will be rewritten as:**

For numerically resolved scales ($|k\Delta x| \leq \pi/2$) where Taylor expansions remain valid, the variable $u$ at neighboring grid points can be approximated as:

**This ensures the convergence of Taylor series expansions, thereby justifying the omission of higher-order terms and enhancing the robustness of our analysis.**

> 18. L303: The approach of this section makes sense and is a nice way of showing that the noise can be controlled. However, it is perhaps worth pointing out that the approach used to do so introduces an arbitrary bias in the direction in which the now piecewise-constant sampling is done, i.e. why choose $u(x)$ instead of $u(x + \Delta x)$? I appreciate that this would not be practicable as a quick experiment, but a better approach might be to always sample upwind, e.g., $u_p(x + \Delta x/2) = u(x)$ when $u > 0$ and $= u(x + \Delta x)$ otherwise.

**We appreciate your suggestion. While the current piecewise-constant sampling introduces directional bias (e.g., choosing $u(x)$ over $u(x+\Delta x)$), this quick experiment demonstrates that unstaggered coupling can effectively suppress the targeted $2\Delta x$ noise. Based on your suggestion,**

we have conducted the experiment using upwind approximation scheme define by Eq. (R1)-(R2).

$$
\begin{cases}
P(u_{i,j}) = P\left(u_{i-\frac{1}{2},j}\right), & if \ \dfrac{u_{i-\frac{1}{2},j} + u_{i+\frac{1}{2},j}}{2} \geq 0 \\[2em]
P(u_{i,j}) = P\left(u_{i+\frac{1}{2},j}\right), & if \ \dfrac{u_{i-\frac{1}{2},j} + u_{i+\frac{1}{2},j}}{2} < 0
\end{cases}
\tag{R1}
$$

and feedback to the dynamics point via:

$$
\begin{cases}
\Delta u^{P}_{i-\frac{1}{2},j} = [P(u_{i,j})]\Delta t, & if \ \dfrac{u_{i-\frac{1}{2},j} + u_{i+\frac{1}{2},j}}{2} \geq 0 \\[2em]
\Delta u^{P}_{i+\frac{1}{2},j} = [P(u_{i,j})]\Delta t, & if \ \dfrac{u_{i-\frac{1}{2},j} + u_{i+\frac{1}{2},j}}{2} < 0
\end{cases}
\tag{R2}
$$

Figure R6 confirms that this methodological alteration does not affect our core conclusions. While we recognize the advantage of implementing an upwind sampling scheme, this improvement would require substantial additional effort to refine – particularly in determining whether to use dynamical-point winds or midpoint-averaged winds as the sampling criterion, along with the rigorous code verification and comprehensive evaluation. Given these requirements, we maintain the current piecewise-constant sampling approach for this study but will add discussion in Section 5, identifying the upwind scheme as a viable near-term option for future implementation (notably simpler than the method in your Comment #2). Please see our response to Comment #21.

[Figure]

Figure R6. Same as Fig.8 in manuscript, but using Eqs. (R1-R2)

19. L310: Please say whether this is also applied for the reverse mapping from the physics point back to the dynamics point.

**Thanks for your suggestion. We will clarify Line 309-314 as:**

In Fig. 1, the physical-point $u$-component can be expanded as $u_p\left(x + \frac{\Delta x}{2}\right) = u(x) + O(\Delta x)$. In Exp_Test, we retain solely the zeroth-order approximation, enforcing $u_p\left(x + \frac{\Delta x}{2}\right) = u(x)$. The

same treatment is applied to the v- component ($v_p\left(y + \frac{\Delta y}{2}\right) = v(y)$), ensuring consistent dynamics-physics coupling for both horizontal velocity components. In correspondence with Eqs. (1)-(2), we now present the modified formulations:

$$\begin{cases} P(u_{i,j}) = P\left(u_{i-\frac{1}{2},j}\right) \\ P(v_{i,j}) = P\left(v_{i,j-\frac{1}{2}}\right) \end{cases} \tag{15}$$

When mapping from physics point back to the dynamics point, we have:

$$\begin{cases} \Delta u^P_{i-\frac{1}{2},j} = P(u_{i,j})\Delta t \\ \Delta v^P_{i,j-\frac{1}{2}} = P(v_{i,j})\Delta t \end{cases} \tag{16}$$

This conformal alignment ensures dynamical and physical processes are computed at coincident grid points, eliminating the need for interpolation-based coupling shown in Eqs. (1) and (2).

> 20. L342: I feel that there is still an interesting level of difference in the spectra over the sea, more than 'remarkable consistency' would suggest.

**We sincerely appreciate your attention to the subtle spectral differences observed over oceanic regions at small scales. The observed small-scale fluctuations appear to originate primarily from islands in our study domain (20°S-20°N, 160°E-120°W). While most of this central Pacific region features flat oceanic surfaces, resolvable islands exist - including Hawaii and numerous small islands east of Australia. As shown in Fig. 10 of the revised manuscript (showing near-surface winds over eastern Australian waters), the Ctrl experiment exhibits clear stripe patterns in low-level winds on the upwind side of islands like Fiji's islands (16-19°S, 177°-180°E) and Vanuatu (14-20°S, 166-171°E), while the Test experiment does not. This observation aligns perfectly with the description in Line 182 (Comment #5) and further validates our study's conclusions regarding the noise generation mechanism.**

**Instead of Line 342-347, a more detailed description will be given in the revised manuscript:**

Over oceanic regions, the homogeneous ocean surfaces produce broadly similar spectral characteristics in both experiments. Although this general spectral agreement suggests that the horizontally homogeneous sea surface lacks the strong physical forcing inhomogeneity required to generate prominent wind stripe patterns, minor differences can still be observed at the smallest resolved scale ($2\Delta x$). While most of the central Pacific study region features homogeneous oceanic surfaces, resolvable islands (e.g., Hawaii and numerous small islands east of Australia) introduce observable heterogeneity. As demonstrated in Fig. 10 (depicting near-surface winds over the eastern Australian waters), the Exp_Ctrl exhibits clear stripe patterns in low-level winds over some of the islands and their upwind regions, such as parts of Fiji (16-19°S, 177°-180°E) and Vanuatu (14-20°S,

166-171°E), whereas Exp_Test shows no such artifacts. This contrast highlights the role of surface inhomogeneity in noise generation.

[Figure]

**Figure 10. 18-hour forecasts of (a)** *u* **- and (b)** *v* **- component for Exp_Ctrl, and (c)** *u* **- and (d)** *v* **- component for Exp_Test, at the lowest model level from CMA - GFS 3.0.**

> 21. L401-409: It is a shame if it is not possible within CMA-GFS to apply the boundary-layer code to wind fields on their native points. This would seem to me to be the preferred solution rather than reintroducing horizontal diffusion. And, as I noted earlier, this is the approach taken within the Met Office's Unified Model.

**We appreciate this valuable suggestion. While implementing the physics scheme directly on the native wind points in CMA-GFS presents significant technical challenges, we recognize—based on Comment #2—that modifying the boundary-layer code remains a viable pathway. We will therefore remove the concluding statement about implementation difficulty from this section and expanded the discussion in the end of Section 5 to explicitly consider adapting the Met Office's Unified Model approach as a potential future improvement. Line 408-409 will be deleted:**

**Line 418-422 will be revised as follows:**

Higher-order horizontal diffusion can serve as a practical remedy for unexplained noise. However, this study has definitively pinpointed the specific sources of wind-field noise in CMA-GFS. We therefore propose the following targeted recommendations:

1. While the piecewise-constant sampling method (Eqs. (15) and (16)) effectively suppresses numerical noise, it may introduce directional biases. A more physically consistent approach would be upwind sampling, where:

$$\begin{cases} u_p\left(x + \dfrac{\Delta x}{2}\right) = u(x) & \text{if } u \geq 0 \\ u_p\left(x + \dfrac{\Delta x}{2}\right) = u(x + \Delta x) & \text{otherswise} \end{cases} \tag{17}$$

This straightforward approach offers immediate operational feasibility and can be rapidly implemented to address the issue. We recommend trial implementation in the operational system followed by comprehensive impact assessment.

2. The strong connection between wind and heat transfer in the boundary layer turbulent diffusion makes it difficult to compute momentum diffusion directly at wind grid points. As demonstrated in Chen et al. (2020), interpolating the diffusivity (rather than prognostic variables) effectively eliminated vertical grid-scale noise in thermodynamic fields — a numerical artifact originally induced by staggered-grid coupling between dynamic and physical processes. Following this approach, we recommend averaging the diffusion coefficient back to the wind points and performing the vertical diffusion on the wind points, thereby avoiding interpolation of prognostic wind variables. Given its demonstrated efficacy in addressing similar discretization challenges in our previous studies (Chen et al., 2020), this approach merits implementation and systematic evaluation.

**Typos/editorial comments**

1. L13: 'dynamical core' is more standard than 'dynamic-core'
2. L21: 'physics-dynamics' is more standard than 'physics-dynamic'
3. L130: word missing from 'therefore systematically these'
4. L243: 'Compared' should be 'Comparing'
5. L280: There is an overbar missing from the $\alpha$
6. L284: The exponent in the middle term, together with the factor $i$, need to be removed

We sincerely appreciate your meticulous attention to detail. All typographical and editorial errors identified will be carefully verified and corrected in the revised manuscript.

Best regards,
Jiong Chen

---

## Author Comment (AC6)

**Von Neumann Analysis for Pure Damping Equation**

For the pure damping equation neglecting advection term:

$$\frac{\partial u}{\partial t} = -\alpha u, \qquad \alpha > 0 \tag{1}$$

where $-\alpha u$ the damping term and $\alpha$ the damping coefficient. Using forward Euler in time, the discretization equation takes the form:

$$\frac{u_j^{n+1} - u_j^n}{\Delta t} = -\alpha_j u_j^n \tag{2}$$

Assume that the numerical solution at $j^{\text{th}}$ point at $n^{\text{th}}$ time step is a single Fourier component of the form:

$$u_j^n = V^n e^{ikj\Delta x} \tag{3}$$

where $k$ is the wavenumber, and $V^n$ the time-dependent complex amplitude at time step $n$. Substituting the Fourier mode from Eq. (3) into Eq. (2), the amplification factor is obtained:

$$G = \frac{V^{n+1}}{V^n} = 1 - \Delta t \alpha_j \tag{4}$$

Equation (4) shows that $G$ is purely real so that the discretization scheme (2) is non-dispersive.

If the damping term is discretized in a staggered manner, similar to the Eq. (1) and (2) in our manuscript, Eq. (2) becomes:

$$\frac{u_j^{n+1} - u_j^n}{\Delta t} = -\frac{1}{2}\left( \alpha_{j-\frac{1}{2}} \frac{u_{j-1}^n + u_j^n}{2} + \alpha_{j+\frac{1}{2}} \frac{u_j^n + u_{j+1}^n}{2} \right) \tag{5}$$

Substituting Eq. (3) into Eq. (5) yields the amplification factor:

$$G = 1 - \frac{\Delta t \bar{\alpha}}{2}\left(1 + \cos(k\Delta x)\right) - i\frac{\Delta t \Delta \alpha}{4}\sin(k\Delta x) \tag{6}$$

where $\bar{\alpha} = \frac{\alpha_{j-\frac{1}{2}} + \alpha_{j+\frac{1}{2}}}{2}, \Delta\alpha = \alpha_{j+\frac{1}{2}} - \alpha_{j-\frac{1}{2}}$.

For waves with wavelengths greater than $2\Delta x$, we have $0 < k\Delta x < \pi$ and therefore $\sin(k\Delta x) \neq 0$. If $\Delta\alpha \neq 0$, the imaginary part of Eq. (6) becomes non-zero. This indicates that the discrete scheme represented by Eq. (5) exhibits dispersive behavior, since the amplification factor now possesses an imaginary component, which introduces a phase shift between wave components. Variations in amplitude are therefore accompanied by changes in phase, leading to frequency-dependent wave propagation speeds.

This analysis leads to the same conclusion as presented in our manuscript.

---

## Author Response (AR1)

**Response to Reviewers' Comments for MS egusphere-2025-2704**

Dear Editor,

We sincerely thank you and the reviewers for the time and insightful comments provided throughout the interactive discussion phase for our manuscript entitled "Stripe Patterns in Wind Forecasts Induced by Physics-Dynamics Coupling on a Staggered Grid in CMA-GFS 3.0" (ID: egusphere-2025-2704). The feedback has been invaluable in helping us improve the quality and clarity of our work.

We are pleased to submit the revised version of our manuscript, which we believe has been significantly strengthened in response to the reviewers' comments. For your convenience, we are providing both a 'marked-up' version with all changes highlighted and a 'clean' version of the revised manuscript with all our changes accepted.

**Overview of Revisions:**

- We have carefully considered and responded to all suggestions from the reviewers, implementing those that enhanced the presentation and clarity of our arguments.
- Following Dr. Añel's comments, the "Code and Data Availability" section has been updated to include the permanent Zenodo DOI for the CMA-GFS 3.0 model code, as previously coordinated.
- In accordance with the journal's guidelines on color accessibility, the line style in Figures 7
  and 9 have been modified to improve distinguishability for readers with color vision
  deficiency (CVD).
- To maintain consistency throughout the manuscript, we have standardized the format for all geographic coordinates to Latitude, Longitude (e.g., Fiji (15–22°S, 175°E–177°W)) instead of Longitude, Latitude.
- Please note that we have updated the author Xueshun Shen as a corresponding author alongside Jiong Chen. This change better reflects his leading role in the research. All authors have approved this modification.

Below, we provide a point-by-point response to all reviewer comments. We are grateful for the constructive and engaging review process and hope that the revisions meet with your approval.

Sincerely, Jiong Chen

**Point-by-Point Responses to Reviewers' Comments**

**Reviewer #1**

We sincerely appreciate Dr. Wood's thorough evaluation and constructive suggestions for our manuscript "Stripe Patterns in Wind Forecasts Induced by Physics-Dynamic Coupling on a Staggered Grid in CMA-GFS 3.0". His insightful comments not only affirm the value of this work but have also significantly helped us refine both the technical rigor and clarity of presentation. In particular, his detailed description of the UK Met Office model's solution to this issue has provided invaluable inspiration, guiding our next steps in adapting and implementing similar improvements in CMA-GFS.

Below we respond point-by-point to his comments, outlining how we have addressed them in the revised manuscript.

**Main point**

The one aspect of the presentation that I think could be improved is for the authors to give the reader better clarity on what is going on and how it leads to the results seen.

In terms of the analysis section 3.4 there are two models of the damping process:

$$\Delta u^\alpha_{i+\frac{1}{2}} = -\Delta t \; \alpha_{i+\frac{1}{2}} \, u_{i+\frac{1}{2}}, \;\; \ldots(A)$$

and

$$\Delta u^{\alpha}_{i+\frac{1}{2}} = -\frac{\Delta t}{2} \left[ \left( \overline{\alpha}_{i+\frac{1}{2}} + \Delta \alpha_{i+\frac{1}{2}} \right) \frac{1}{2} \left( u_{i+\frac{3}{2}} + u_{i+\frac{1}{2}} \right) + \left( \overline{\alpha}_{i+\frac{1}{2}} - \Delta \alpha_{i+\frac{1}{2}} \right) \frac{1}{2} \left( u_{i+\frac{1}{2}} + u_{i-\frac{1}{2}} \right) \right]. \quad \dots (B)$$

There are two steps that lead to noise being induced by process B.

The first step is common to both A and B. It is perhaps clear to many readers but I think it would be worth being explicit that applying either A or B to even (or perhaps especially) an initially constant field will induce a variation in that field that reflects the variation of  $\alpha$ . Such an effect is clear from the results of the control experiment in Fig 7e where the initially uniform wind is damped to zero at one point but remains almost at its initial value upstream of that point – the horizontal variation of  $u_*$  is reflected in the wind field.

The next step in the argument is that for process B, the combination of horizontal variation in  $\alpha$  and averaging of fields back and forth leads to a dispersive error in how the wind field evolves that is not seen for process A (for which the impact of the damping coefficient is purely local). That this is the case can be seen by rewriting process B as:

$$\Delta u_{i+\frac{1}{2}}^{\alpha} = -\frac{\overline{\alpha}_{i+\frac{1}{2}}\Delta t}{4} \left( u_{i+\frac{3}{2}} + 2u_{i+\frac{1}{2}} + u_{i-\frac{1}{2}} \right) - \frac{\Delta \alpha_{i+\frac{1}{2}}\Delta t}{4} \left( u_{i+\frac{3}{2}} - u_{i-\frac{1}{2}} \right). \dots (\mathcal{C})$$

In this form two things become apparent:

- 1. The first term is perhaps not surprising. It has the form of a second-order accurate horizontal diffusion scheme. Any  $2\Delta x$  mode, that emerges by whatever route, is invisible to this term and so will not be damped at all. Other modes will also be impacted by the smoothing inherent in the 1-2-1 operator. (In contrast, process A will damp all modes equally as efficiently.)
- 2. More interestingly perhaps, the second term can be interpreted as a second-order centred-difference advection scheme where the advecting "velocity" is  $\Delta x \Delta \alpha/2$ . This term is therefore not damping at all. It is also the source of the dispersion issue: When applied to a field with a discontinuous field such a scheme is well known to create upstream propagating noise. Indeed, Fig 3.7a of Durran's second edition of Numerical Methods for Fluid Dynamics is very reminiscent of the form of the result of process B shown in Fig 7e of the present work.

None of this says a lot more than the authors already present in their work but I feel that adding a brief discussion of the form (C) and making the analogy with centred advection (and its dispersion error and associated noise) might, for some readers, give a bit more insight into what is happening.

Response: Dr. Wood's comments are highly insightful and have prompted us to more deeply consider the origin of the dispersion error demonstrated in our study. Eq. (C) is particularly interesting as it further clarifies that the true dispersion error stems from the second term in the equation. In response to his suggestions, we propose the following modifications:

1. The variables configuration in staggered coupling using wave analysis is demonstrated in Figure R1. This configuration maintains consistent subscript notation with the formulas in Durran's book (Section 3.3).

Figure R1. The configuration of second model of damping process shown as Eq. (10) in the following revised version

- 2. This study focuses solely on the numerical coupling impact of local friction terms, not advection discretization—a distinction clarified in the revised manuscript. Our idealized experiments demonstrate that spatially varying friction induces non-uniform u distributions (e.g., spiked structures), whose differential responses depend on the discretization scheme. As Durran's description in Section 3.3 of his book, central differencing of advection introduces dispersion artifacts near discontinuities, which is also why we employ first-order upwind scheme for advection in our idealized experiments to isolate the physics-dynamic coupling issue in this study.
- 3. As noted in his Comment #13, our analysis in the manuscript considered only one component of the averaging process—specifically, the physics-to-dynamics interpolation (Eq.2) without accounting for the dynamics-to-physics averaging (Eq. 1). Following his suggestion, we confirm this revised derivation does yield additional terms with modified

coefficients, as his anticipated (See Eq. (11), here  $\Delta \alpha_j = \alpha_{j+\frac{1}{2}} - \alpha_{j-\frac{1}{2}}$  based on his comment #16).

4. Following his suggestion, we have incorporated a discussion of Equation C, which provides deeper insight into the underlying mechanism—benefiting both readers and our own understanding of this phenomenon's root cause.

To reflect these, we have thoroughly revised Section 3.4. Please see the lines 279–340 in the marked-up (traceable) manuscript version showing the changes made and lines 271–319 in the clean copy with all our changes accepted.

**Minor comments**

(Lnn refers to line number nn.)

1. L13: I am not convinced that the comment about the absence of noise in the static fields is quite correct – see my comment below about L177. I would suggest instead saying something like 'the structure of the static fields in not consistent with the amplitude of the  $2\Delta x$  noise if that noise were forced locally by the static fields'?

Response: We appreciate Dr. Wood's suggestion and will adopt the following more rigorous description in our revised manuscript. The changes can be found in the traceable version (Lines 13–18) and in the clean version (Lines 13–16).

2. L80-83: The UK Met Office model does not follow this approach. It averages the winds to the cell centre but only uses these averaged winds to evaluate the boundary layer diffusion coefficient (the eddy diffusivity). It then averages the diffusion coefficient back to the wind points and performs the vertical diffusion at the wind points. This is the approach that I would recommend in solving the problem presented in this paper if your infrastructure can support such an approach.

Response: We sincerely appreciate Dr. Wood for sharing how the UK Met Office model addresses this issue. Based on the analysis presented in this study, we are confident that this approach will effectively resolve the noise problem. We will implement and evaluate corresponding improvements in CMA-GFS. Please also refer to our response to Comment #21 for additional clarification on this point.

3. L156-157: It would be good somewhere to comment on the stability of the model, i.e. is the amplitude of the stripes approximately constant in that they appear and remain approximately unchanged, or do they grow in time?

Response: As shown in Figure 3, the amplitude of the stripes remains approximately unchanged with forecast days despite pronounced diurnal variations. Therefore, we have revised the manuscript accordingly (traceable: Line 162–164; clean: Line 160–161).

4. L177: Without further evidence to support this, I think 'complete absence' is too strong. It is clear that there is not the same visual level of noise at the  $2\Delta x$  scale as in Fig 2 but that is not the same as there being a complete absence. And given that horizontal averaging of a field cannot create a  $2\Delta x$  component then I think that the heart of the later argument lies in there being some forcing of such a component by the physics. It would be interesting to present a spectral analysis of these fields in the same way as Fig 9

Response: We sincerely appreciate Dr. Wood's attention to this detail. Spectral analysis in Fig. R2 demonstrates small-scale fluctuations in both standard deviation of sub-grid orography and roughness length, which are also visually evident in Fig. R3. These results demonstrate the existence of small-scale fluctuations in the static fields. However, as shown in Fig. 5 of our manuscript, these static fields do not exhibit the stripe patterns observed in the wind field distribution, thereby excluding them as direct sources of the stripe noise. Their spectral characteristics differ from the wind field noise: the static fields' energy spectra decay at smaller scales (Fig. R2), whereas the Ctrl\_EA wind field spectra intensify at these scales (Fig. 9). This contrast suggests that small-scale static inhomogeneities are probably not the primary direct source of the observed wind stripes.

Critically, static field inhomogeneity (particularly the prominent spikes in Fig. R3) serves as triggering factors for the noise through wave dispersion effects - as discussed in Sections 3.3 and 3.4. Based on these considerations, the term 'complete absence' is scientifically inappropriate. To align with the overall logic of our analysis, we have revised the sentence (traceable: Line 184–188; clean: Line 181–182).

Figure R2. PSD of static field over the East Asia (70°-145°E, 10°-65°N): roughness length along *x*-direction: solid line; roughness length along *y*-direction: dashed line; orography standard deviation along *x*-direction: dash-dot line; orography standard deviation along *y*-direction dotted line

Figure R3. The roughness length along 100°E

5. L182: It would be useful to the reader to give an indicative latitude and longitude for where the islands are.

Response: We appreciate this suggestion. The islands discussed in our study are located at approximately 93°E, 11–13°N. This information will be added to the revised manuscript (traceable: Line 191; clean: Line 186).

6. L209, section 3.3: I think here one is looking for the simplest experimental set-up that shows the noisy behaviour. Given the theoretical model used in section 3.4 I would have thought that it would be best in this section to match as closely as possible that theoretical model, i.e. use a constant advecting wind (thereby losing the complication of nonlinearity) and use a constant eddy viscosity in the vertical (thereby losing the complication of nonzero values of dK/dz. If the hypothesis is correct then the equivalent figures to Figs 7a, b and e should be very similar.

Response: We sincerely appreciate this insightful suggestion. As Dr. Wood correctly noted, both linear advection and constant eddy viscosity (K) successfully reproduce the  $2\Delta x$  waves near surface friction point (Fig. R4). However, our analysis reveals that nonlinear advection amplifies these  $2\Delta x$  fluctuations, which may explain why the stripe noise is so obvious in the CMA-GFS's wind forecasts. Additionally, larger K values enhance the friction impact on upper boundary layer (Fig. R4, Right panel).

To balance realism with simplicity, we retain nonlinear advection (to best replicate the full model's behavior) while adopting constant K for idealized experiments. This configuration (Fig. R4, middle panel) has been selected for consistency. Fig. 7 and Eq. (5) has been updated accordingly (traceable: Line 225; clean: Line 220).

Figure R4. Left: Linear advection with c=10m s $^{-1}$ , but retain height-dependent K; Middle: nonlinear advection with constant K=10 m $^2$  s $^{-1}$ ; Right: nonlinear advection with constant K=50 m $^2$  s $^{-1}$ .

7. L216: If the authors do retain a height varying *K* then it is probably worth saying that this is an example profile for the purposes of the idealised model rather than the profile used in the full model.

Response: Thank Dr. Wood for this suggestion. Consistent with our stated experimental design (Response #6), we have employed the constant eddy viscosity (K) configuration for the idealized simulations.

8. Equation (5): I believe there is a missing factor of  $u_*$ .

Response: We thank Dr. Wood for prompting deeper consideration of this issue. As shown in Fig. R5, u\*-dependent K becomes highly localized (peaking at the friction point). With a constant 500-m boundary layer height, our setup approximates an unstable boundary layer in all x points. Since unstable boundary layer involves both thermal-driven turbulence effects (absent here) and wind shear-driven turbulence, using  $u_*$ -dependent K would demand simultaneous heat flux modeling. Our constant-K choice (as detailed in Response #6) avoids this complexity while maintaining physical consistency with the prescribed constant boundary layer depth.

Figure R5. Linear advection with u\*-dependent K

9. L220/236: To save the reader having to work it out for themselves, it would be good to state that the time step is 300s (if I have worked it out correctly!).

Response: We am very sorry we forgot to state it. The time step is 300s and it has been added in Table 1. Thank Dr. Wood very much for his careful attention to this detail.

10. L241: It would be useful to say whether the noise grows or is stable.

Response: Thank Dr. Wood for this important clarification. As shown in Fig. 7b, the 2Δx wave amplitude remains nearly constant throughout the integration period, mirroring the behavior in CMA-GFS. This stability has bedn explicitly stated in the revised manuscript (traceable: Line 251–252; clean: Line 246–247).

11. L252, Fig 7: It would be worth stating in the caption that 'x-grid' means the grid point number not a distance.

Response: We appreciate this clarification. The caption has been revised (traceable: Line 264; clean: Line 258).

12. L255-256: I am not convinced that the words used for either source of 'noise' are correct. Earlier it has been stated that there is a complete absence of noise in the surface fields of the full model! The surface forcing is what it is – it is the amplification, or exposure (through dispersion), of whatever  $2\Delta x$  component that is at question. Also, I think it is debatable whether that averaging is an inconsistency; I would suggest that it is a discretization choice (albeit perhaps not a good choice!).

Response: We appreciate these nuanced observations. The wording should be more precisely characterize the mechanisms:

1. Surface effects: The original text suggested surface forcing directly creates noise, which was

- inaccurate. Instead, surface roughness variations (Fig. R3) interact with the model's numerical dispersion, amplifying pre-existing  $2\Delta x$  waves.
- Grid staggering: The term 'inconsistent' was misleading. The issue arises because advection and diffusion use different grid arrangements (staggered vs. non-staggered), which can artificially enhance certain wave modes.

Therefore this sentence has been modified (traceable: Line 267–271; clean: Line 261–264).

13. L272 and following lines: The authors have included one part of the averaging process, from the physics points (cell centres) to the velocity points (cell faces). They have then used Taylor series expansions to estimate the winds at the cell centres. It would be better (and no more difficult) to explicitly include the averaging of the winds to the cell centres and then use Taylor series expansions to obtain estimates for  $u(x \pm \Delta x)$  in terms of u(x). This has the effect of changing, in Eq. (10), the denominator 8 to 4, and the denominator 48 to 12. Although this is only a minor change in practice, it better reflects what the model of section 3.3 actually does, and perhaps more usefully, the terms that are proportional to  $\Delta \alpha$  are proportional to  $u' + \Delta x^2 u'''/6$  which are the terms that arise in a second-order centred advection scheme (as expected from the analysis suggested in the section Main Point).

Response: We appreciate Dr. Wood's insightful suggestion regarding the averaging process. As noted in our response to the Main Point comment, we have revised the derivation in Section 3.3 to explicitly include the averaging of winds to cell centers before applying Taylor series expansions. Following the comment #16, this adjustment now modifies the coefficients in Eq. (11) (originally Eq.(10)) to 4, 4, and 24, which more accurately represents the model's numerical formulation.

14. L275: Strictly, Eq. (10) is only the solution to Eq. (8) if  $\overline{\alpha}$  and  $\Delta \alpha$  are taken as independent of x which they cannot both be over a periodic domain. You could either rewrite (8) in terms of  $\overline{\alpha}$  and  $\Delta \alpha$  and then postulate a different problem that has those coefficients constant, or simply say that the solution is approximate and only holds locally.

Response: We agree with this observation. In Eq. (11) [modification of (10)],  $\overline{\alpha}_j$  and  $\Delta \alpha_j$  are evaluated at a given grid point  $x_j$ .

15. L277: Related to the above point, I think here (and everywhere they are used)  $\overline{\alpha}$  and  $\Delta \alpha$  should be written as  $\overline{\alpha}_{i+1/2}$  and  $\Delta \alpha_{i+1/2}$  to make their horizontal dependency clear.

Response: We fully agree with this suggestion. As illustrated in Fig. R1, we have implemented the proposed changes (traceable: Line 308; clean: Line 290).

16. L277: It would be much better to not include the factor of 1/2 in the definition of  $\Delta \alpha$  but carry that explicitly where  $\Delta \alpha$  is used. Otherwise the definition is not consistent with the use of  $\Delta$  elsewhere, e.g., consider what happens if  $\alpha \equiv x$ , we would then end up with  $\Delta x = \Delta x/2!$

Response: We agree and have removed the factor of 1/2 from the definition of  $\Delta\alpha_j$ , carrying it explicitly in Eq.(11) (traceable: Line 308; clean: Line 290). This correction aligns with standard notation and avoids scaling inconsistencies.

17. L280: It would be worth reminding the reader somewhere that the Taylor series expansion is only valid for small values of  $k\Delta x$ . Otherwise, when  $k\Delta x = \pi$ , as it can do, the solution (both (10) and (11)) would be predicted to grow unlimitedly since  $\pi^2/8 - 1 > 0$ .

Response: Thank Dr. Wood for this critical insight. As now clarified before Eq. (10), our Taylor expansions are strictly valid for numerically resolved scales ( $|k\Delta x|$  << 1) (traceable: Line 301; clean: Line 285). This ensures the convergence of Taylor series expansions, thereby justifying the omission of higher-order terms and enhancing the robustness of our analysis.

18. L303: The approach of this section makes sense and is a nice way of showing that the noise can be controlled. However, it is perhaps worth pointing out that the approach used to do so introduces an arbitrary bias in the direction in which the now piecewise-constant sampling is done, i.e. why choose u(x) instead of  $u(x + \Delta x)$ ? I appreciate that this would not be practicable as a quick experiment, but a better approach might be to always sample upwind, e.g.,  $u_p(x + \Delta x/2) = u(x)$  when u > 0 and  $u(x + \Delta x)$  otherwise.

Response: We appreciate Dr. Wood's suggestion. While the current piecewise-constant sampling introduces directional bias (e.g., choosing u(x) over  $u(x+\Delta x)$ ), this quick experiment demonstrates that unstaggered coupling can effectively suppress the targeted  $2\Delta x$  noise. Based on this suggestion, we have conducted the experiment using upwind approximation scheme define by Eq. (R1)-(R2).

$$\begin{cases} P(u_{i,j}) = P\left(u_{i-\frac{1}{2},j}\right), & if \frac{u_{i-\frac{1}{2},j} + u_{i+\frac{1}{2},j}}{2} \ge 0\\ P(u_{i,j}) = P\left(u_{i+\frac{1}{2},j}\right), & if \frac{u_{i-\frac{1}{2},j} + u_{i+\frac{1}{2},j}}{2}

Figure R6. Same as Fig.8 in manuscript, but using Eqs. (R1-R2)

19. L310: Please say whether this is also applied for the reverse mapping from the physics point back to the dynamics point.

Response: Thanks for this suggestion. The same principle applies to the reverse mapping when physics feedback to the dynamics. We have clarified this in the revised manuscript (traceable: Line 353–362; clean: Line 332–341).

20. L342: I feel that there is still an interesting level of difference in the spectra over the sea, more than 'remarkable consistency' would suggest.

Response: We sincerely appreciate Dr. Wood's attention to the subtle spectral differences observed over oceanic regions at small scales. The observed small-scale fluctuations appear to originate primarily from islands in our study domain (20°S-20°N, 160°E-120°W). While most of this central Pacific region features flat oceanic surfaces, resolvable islands exist - including Hawaii and numerous small islands east of Australia. As shown in Fig. 10 of the revised manuscript (showing near-surface winds over eastern Australian waters), the Ctrl experiment exhibits clear stripe patterns in low-level winds on the upwind side of islands like Fiji's islands (16-19°S, 177°-180°E) and Vanuatu (14-20°S, 166-171°E), while the Test experiment does not. This observation aligns perfectly with the description in section 3.1 and further validates our study's conclusions regarding the noise generation mechanism.

Instead of Line 342-347 in our original manuscript, a more detailed description has been given in the revised version (traceable: Line 389–405; clean: Line 367–378).

21. L401-409: It is a shame if it is not possible within CMA-GFS to apply the boundary-layer code to wind fields on their native points. This would seem to me to be the preferred solution rather than reintroducing horizontal diffusion. And, as I noted earlier, this is the approach taken within the Met Office's Unified Model.

Response: We appreciate this valuable suggestion. While implementing the physics scheme directly on the native wind points in CMA-GFS presents significant technical challenges, we recognize—based on Comment #2—that modifying the boundary-layer code remains a viable

pathway. We will therefore remove the concluding statement about implementation difficulty from this section and expanded the discussion in the end of Section 5 to explicitly consider adapting the Met Office's Unified Model approach as a potential future improvement. This sentence has been deleted and a more detailed discussion has been given (traceable: Line 353–362 and Line 479–500; clean: Line 332–341 and Line 451–469).

**Typos/editorial comments**

- 1. L13: 'dynamical core' is more standard than 'dynamic-core'
- 2. L21: 'physics-dynamics' is more standard than 'physics-dynamic'
- 3. L130: word missing from 'therefore systematically these'
- 4. L243: 'Compared' should be 'Comparing'
- 5. L280: There is an overbar missing from the  $\alpha$
- 6. L284: The exponent in the middle term, together with the factor i, need to be removed

Response: We sincerely appreciate Dr. Wood's meticulous attention to detail. All typographical and editorial errors identified has been carefully verified and corrected in the revised manuscript.

1. traceable: Line 15; clean: Line 15

2. traceable: Line 25; clean: Line 23

3. The word "investigated" has been added (traceable: Line 134; clean: Line 132).

4. traceable: Line 254; clean: Line 2495. traceable: Line 313; clean: Line 293

6. traceable: Line 317; clean: Line 297

**Reviewer #2**

We appreciate Dr. Santos very much for his thoughtful review and for his encouraging comments on our manuscript.

Regarding Dr. Santos' two specific comments, we respond as follows:

1. Comment on Equation (7): First, the equation (7) in the manuscript is not the analytic solution to (6) if  $\alpha$  is spatially varying. An example is given in the attachment. The effect of the spatially-varying  $\alpha$  is simply to modulate the amplitude of the wave through inhomogeneous damping, so I do not think this causes problems for the paper's main argument. I think it would be fine to omit the analytic solution in (7) entirely, since the important point in this section is that using half-grid-scale offsets introduces extra dispersion not present in other discretizations.

Response: Dr. Santos is absolutely correct that Equation (7) is not an exact analytical solution to Equation (6) when  $\alpha$  varies spatially. Following Dr. Wood's comments, our revised discussion now focuses on the discrete solution of Equation (6). Therefore, we have removed the original analytical solution (Equation 7) and its related discussion from the manuscript (traceable: Line 279–282; clean: Line 271–271).

2. **Comment on the term "discontinuous" (Line 299):** From this paper, I don't see that you necessarily need surface friction to be discontinuous at some particular point to produce these numerical artifacts; rather, surface friction only needs significant variability at the grid scale.

Response: We thank Dr. Santos for this meticulous observation. He is right that the presence of numerical artifacts does not require surface friction to be strictly discontinuous, but only to exhibit significant grid-scale variability. Accordingly, we have replaced the word "discontinuous" with "non-uniform" (traceable: Line 343; clean: Line 321–322).

---

## Author Response (AR2)

Dear Dr. Caldwell,

Thank you very much for handling the review of our manuscript, "Stripe Patterns in Wind Forecasts Induced by Physics-Dynamics Coupling on a Staggered Grid in CMA-GFS 3.0" (Manuscript ID: egusphere-2025-2704). We are pleased to hear that the manuscript has been accepted pending minor revisions. We have carefully considered all your suggestions and have revised the manuscript accordingly. The changes are detailed below.

1. L54-57: I found description of the Lorenz (L) and Charney-Phillips (CP) grids to be insufficient. For example, if thermodynamic and dynamic variables are at half levels on the L grid, what's at full levels? I think the authors are missing description of vertical winds.

Response: We agree and thank you for this suggestion. We have revised the text to explicitly state the location of the vertical velocity and to clarify the variables at the full-levels. Please see the lines 54-60 in the revised manuscript.

2. I found it odd that the authors insert a "{" symbol to the left of systems of equations. If this is normal in Europe disregard my comment.

Response: We thank you for this careful observation. We have checked the formatting against published articles in GMD and, accordingly, have removed the left curly brace '{' from equations (1-2) and (14-16) to align with the journal's standard convention.

3. Similarly, I think Eq 5 should start with " $K(z) = \{$ " rather than having K(z) on the upper condition.

Response: We agree with you. We have revised Equation 5 to place 'K(z) =' before the opening curly brace, as you suggested.

4. eq 17: "otherwise" is misspelled.

Response: We have revised this typo in Eq. 17. Thank you very much.

5. line 50: Need "s" after "model" in "In NWP model, ..."

Response: Thank you very much for this careful observation. We have revised this typo. Please see Line 48.

In accordance with the 'Notification to the Authors', we have ensured that Figure 4 is successfully embedded in the PDF and that all figures have been verified—using the Coblis simulator—to meet accessibility standards for readers with color vision deficiencies.

Additionally, we have corrected two minor errors in the text.

(1) We have corrected the equation label that was mistakenly identified as '17' to the correct number '16'.

(2) We have also adjusted the format of the longitude and latitude at Line 148 to ensure consistency throughout the manuscript.

Thank you for your time and consideration.

Sincerely, Jiong Chen